# BiSLS/SPS: Auto-tune Step Sizes for Stable Bi-level Optimization

**Chen Fan[1], Gaspard Choné-Ducasse[2], Mark Schmidt[1,3], and Christos Thrampoulidis[1]**

[1]University of British Columbia
[2]Ecole Normale Supérieure
[3]Canada CIFAR AI Chair (Amii)

## Abstract

The popularity of bi-level optimization (BO) in deep learning has spurred a growing interest in studying gradient-based BO algorithms. However, existing algorithms involve two coupled learning rates that can be affected by approximation errors when computing hypergradients, making careful fine-tuning necessary to ensure fast convergence. To alleviate this issue, we investigate the use of recently proposed adaptive step-size methods, namely stochastic line search (SLS) and stochastic Polyak step size (SPS), for computing both the upper and lower-level learning rates. First, we revisit the use of SLS and SPS in single-level optimization without the additional interpolation condition that is typically assumed in prior works. For such settings, we investigate new variants of SLS and SPS that improve upon existing suggestions in the literature and are simpler to implement. Importantly, these two variants can be seen as special instances of general family of methods with an envelope-type step-size. This unified envelope strategy allows for the extension of the algorithms and their convergence guarantees to BO settings. Finally, our extensive experiments demonstrate that the new algorithms, which are available in both SGD and Adam versions, can find large learning rates with minimal tuning and converge faster than corresponding vanilla SGD or Adam BO algorithms that require fine-tuning.

## 1 Introduction

Bi-level optimization (BO) has found its applications in various fields of machine learning such as hyperparameter optimization [14, 17, 30, 40], adversarial training [51], data distillation [2, 53], neural architecture search [28, 39], neural-network pruning [52], and meta-learning [13, 37, 11]. Specifically, BO is used widely for problems that exhibit a hierarchical structure of the following form:

$$\min_{x \in X} F(x) = \mathbb{E}_\phi[f(x, y^*(x); \phi)] \qquad \text{s.t.} \qquad y^*(x) = \operatorname*{argmin}_{y \in Y} \mathbb{E}_\psi[g(x, y; \psi)]. \tag{1}$$

Here, the solution to the lower-level objective $g$ becomes the input to the upper-level objective $f$, and in (1) the upper-level variable $x$ is fixed when optimizing the lower-level variable $y$. To solve such bi-level problems using gradient-based methods requires computing the hypergradient of $F$, which based on the chain rule is given as [15]:

$$\nabla F(x) = \nabla_x f(x, y^*(x)) - \nabla^2_{xy} g(x, y^*(x))[\nabla^2_{yy} g(x, y^*(x))]^{-1} \nabla_y f(x, y^*(x)). \tag{2}$$

In practice, the closed-form solution $y^*(x)$ can be difficult to obtain, and one strategy is to run a few steps of (stochastic) gradient descent on $g$ with respect to $y$ to get an approximation $\bar{y}$, and

use $\bar{y}$ in places of $y^*(x)$. We denote the stochastic hypergradient based on $\bar{y}$ as $h_f(x, \bar{y})$ and the stochastic gradient of $g$ with respect to $y$ as $h_g$. This leads to a general gradient-based framework for solving bi-level optimization [15, 19, 4]. At each iteration $k$, run T (can be one or more) steps of SGD on $y$ with a step size $\beta$, $y^{k,t+1} = y^{k,t} - \beta h_g^{k,t}$, then run one step on $x$ using the approximated hypergradient:

$$x^{k+1} = x^k - \alpha h_f(x^k, y^{k+1}), \quad \text{where} \quad y^{k+1} = y^{k,T}. \tag{3}$$

Based on this framework, a series of stochastic algorithms have been developed to achieve the optimal or near-optimal rate of their deterministic counterparts [7, 8]. These algorithms can be broadly divided into single-loop ($T = 1$) or double-loop ($T > 1$) categories [23].

Unlike minimizing the single-level finite-sum (convex) problem

$$F(x) := \min_{x \in \mathcal{C}} \frac{1}{N} \sum_{i=1}^{N} f_i(x), \tag{4}$$

where only one learning rate is involved when using SGD, bi-level optimization involves tuning both the lower and upper-level learning rates ($\beta$ and $\alpha$ respectively). This poses a significant challenge due to the potential correlation between these learning rates [19]. Thus, as

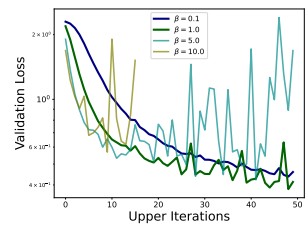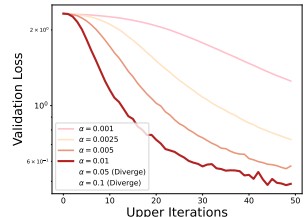

Figure 1: Results based on hyper-representation learning task (see Sec 4 for details). Validation loss against upper-level iterations for different values of $\beta$ (left, $\alpha = 0.005$) and $\alpha$ (right, $\beta = 0.01$). Unless carefully tuned, vanilla SGD-based methods for BO are very unstable.

observed in Figure 1, algorithm divergence can occur when either $\alpha$ or $\beta$ is large. While there is considerable literature on achieving faster rates in bi-level optimization [24, 5, 7, 8], only a few studies have focused on stabilizing its training and automating the tuning of $\alpha$ and $\beta$.

This work addresses the question: **Is it possible to utilize large $\alpha$ and $\beta$ without manual tuning?** In doing so, we explore the use of stochastic adaptive-step size methods, namely stochastic Polyak step size (SPS) and stochastic line search (SLS), which utilize gradient information to adjust the learning rate at each iteration [44, 29]. These methods have been demonstrated to perform well in interpolation settings with strong convergence guarantees [44, 29]. However, applying them to bi-level optimization (BO) introduces significant challenges, as follows. BO requires tuning two correlated learning rates (for lower and upper-level). The bias in the stochastic approximation of the hypergradient complicates the practical performance and convergence analysis of SLS and SPS. Other algorithmic challenges arise for both algorithms. For SLS, verifying the stochastic Armijo condition at the upper-level involves evaluating the objective at a new $(x, y^*(x))$ pair, while $y^*(x)$ is only approximately known; For SPS, most existing variants guarantee good performance only in interpolating settings, which are typically not satisfied for the upper-level objective in BO [22]. Before presenting our solutions to the challenges above in Sec 2, we first review the most closely related literature.

## 1.1 Related Work

**Gradient-Based Bi-level Optimization** Penalty or gradient-based approaches have been used for solving bi-level optimization problems [10, 45, 21]. Here we focus our discussions on stochastic gradient-based methods as they are closely related to this work. For double-loop algorithms, an early work (BSA) by Ghadimi and Wang [15] has derived the sample complexity of $\phi$ in achieving an $\epsilon$-stationary point to be $\mathcal{O}(\epsilon^{-2})$, but require the number of lower-level steps to satisfy $T \sim \mathcal{O}(\epsilon^{-1})$. Using a warm start strategy (stocBiO), Ji et al. [22] removed this requirement on $T$. However, to achieve the same sample complexity, the batch size of stocBiO grows as $\mathcal{O}(\epsilon^{-1})$. Chen et al. [4] removed both requirements on $T$ and batch size by using the smoothness properties of $y^*(x)$ and setting the step sizes $\alpha$ and $\beta$ at the same scale. For single-loop algorithms, a pioneering work by Hong et al. [19] gave a sample complexity of $\mathcal{O}(\epsilon^{-2.5})$, provided $\alpha$ and $\beta$ are on two different scales (TTSA). By making corrections to the $y$ variable update (STABLE), Chen et al. [5] improved the rate to $\mathcal{O}(\epsilon^{-2})$. However, extra matrix projections required by STABLE can incur high computation cost [5, 4]. By incorporating momentum into the updates of $x$ and $y$ (SUSTAIN), Khanduri et al.

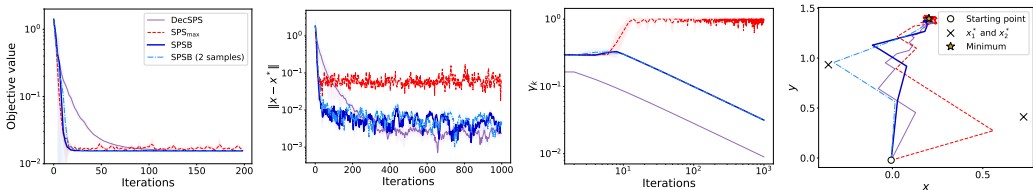

Figure 2: Experiments on quadratic functions adapted from [29]. The objective is the sum of two-dimensional functions $f_i = \frac{1}{2}(x - x_i^*)^T H_i(x - x_i^*)$, where $H_i$ is positive definite and $i = 1, 2$ (see Appendix B for more details). From left to right, we show: the objective value, distance to optimum, step size, and iterate trajectories.

[24] further improved the rate to $\mathcal{O}(\epsilon^{-1.5})$ [6]. Besides these single or double-loop algorithms, a series of works have drawn ideas from variance reduction to achieve faster convergence rates for BO. For example, Yang et al. [49] designed the VRBO algorithm based on SPIDER [12]. Dagréou et al. [7, 8] designed the SABA and SRBA algorithms based on SAGA and SARAH respectively, and demonstrate that they can achieve the optimal rate of $\mathcal{O}(\epsilon^{-1})$ [9, 35]. Huang et al. [20] proposes to use Adam-type step sizes in BO. However, it introduces three sequences of learning rates $(\alpha_k, \beta_k, \eta_k)$ that require tuning, which limits its practical usage. To our knowledge, none of these works have explicitly addressed the fundamental problem of how to select $\alpha$ and $\beta$ in bi-level optimization. In this work, we focus on the alternating SGD framework (T can be 1 or larger), and design efficient algorithms that find large $\alpha$ and $\beta$ without tuning, while ensuring the stability of training.

**Adaptive Step Size** Adaptive step-size such as Adam has found great success in modern machine learning, and different variants have been proposed [25, 38, 47, 31, 32]. Here, we limit our discussions on two adaptive step sizes that are most relevant to this work. The Armijo line search is a classic way for finding step sizes for gradient descent [48]. Vaswani et al. [44] extends it to the stochastic setting (SLS) and demonstrates that the algorithm works well with minimal tuning required under interpolation, where the model fits the data perfectly. This method is adaptive to local smoothness of the objective, which is typically difficult to predict a priori. However, the theoretical guarantee of SLS in the non-interpolating regime is lacking. In fact, the results in Figure 3 suggest that SLS can perform poorly for convex losses when interpolation is not satisfied. Besides SLS, another adaptive method derived from the Polyak step size is proposed by Loizou et al. [29] with the name stochastic Polyak step size (SPS). Loizou et al. [29] further places an upper bound on the step size resulting in the $\text{SPS}_{\max}$ variant. Similar to SLS, the algorithm performs well when the model is over-parametrized. Without interpolation, the algorithm converges to a neighborhood of the solution whose size depends on this upper bound.

---

**Algorithm 1** BiSLS-Adam/SGD

---

**Input:** $x^0, y^0, K, T, \delta, \alpha_{b,0}, \beta_{b,0}, \eta, w \in (0,1)$
**Output:** $x$
1: **for** $k = 0, 1, \ldots, K - 1$ **do**
2:     $y^{k,0} = y^k$
3:     **for** $t = 0, 1, \ldots, T - 1$ **do**
4:         $\beta_{b,k}^t \leftarrow \text{reset}(\beta, \beta_{b,0}, \eta, \text{opt})$   ▷ see Algorithm 2
5:         $\beta \leftarrow$ line-search based on (8) starting from $\beta_{b,k}^t$
6:         $y^{k,t+1} = y^{k,t} - \beta\, h_g^{k,t}$,
7:     **end for**
8:     $y^{k+1} = y^{k,T-1}; \hat{x}^k = x^k; \hat{y}^{k+1} = y^{k+1}$
9:     $\alpha \leftarrow \text{reset}(\alpha, \alpha_{b,0}, \eta, \text{opt})$
10:     **while** (14) based on $(\hat{x}^k, \hat{y}^{k+1}, \alpha, \delta)$ does not hold. **do**
11:         $\alpha = \alpha * w$
12:         $\hat{x}^k = x^k - \alpha\, h_f(x^k, y^{k+1})$ or
13:         $\hat{x}^k = x^k - \alpha A_k^{-1} h_f(x^k, y^{k+1})$
14:         $\hat{y}^{k+1} = y^{k+1} - \beta\, h_g(\hat{x}^k, y^{k+1})$
15:     **end while**
16:     $x^{k+1} = x^k - \alpha\, h_f^k$
17: **end for**

---

In a later work, Orvieto et al. [36] make the SPS converge to the exact solution by ensuring the step size and its upper bound are both non-increasing (DecSPS ). However, enforcing monotonicity may result in the step size being smaller than decaying-step SGD and losing the adaptive features of SPS (see Figure 2, 3). In this work, we propose new versions of SLS and SPS that do not require monotonicity and extend them into the alternating SGD bi-level optimization framework (3).

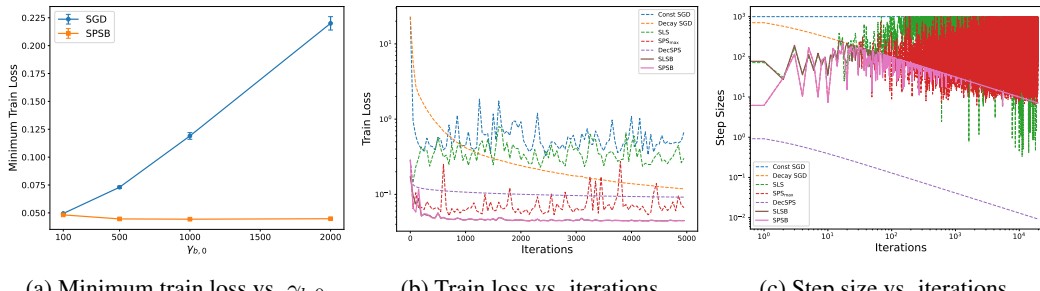

| (a) Minimum train loss vs. $\gamma_{b,0}$ | (b) Train loss vs. iterations | (c) Step size vs. iterations |

Figure 3: Binary linear classification on w8a dataset using logistic loss [3]. (a) Minimum train loss of decaying-step SGD and SPSB for different $\gamma_{b,0}$'s. (b)(c) Train loss and step size against iterations, respectively. We choose $c = 1$ and $\bar{c} = 1$ for $\text{SPS}_{\max}$ and SLS respectively; $c_k = \sqrt{k+1}$ for DecSPS ; $c_k = 1$ and $\gamma_{b,k} = \frac{\gamma_{b,0}}{\sqrt{k+1}}$ for SPSB ; $\bar{c} = 0.1$ and $\gamma_{b,k} = \frac{\gamma_{b,0}}{\sqrt{k+1}}$ for SLSB ; $\gamma_{b,k} = \frac{\gamma_{b,0}}{\sqrt{k+1}}$ for decaying-step SGD. For (b) and (c), we set $\gamma_{b,0} = 1000$ for all algorithms.

## 2 Summary of Contributions

We discuss our main contributions in this section, which is organized as follows. First, we discuss our variants of SPS and SLS, and unify them under the umbrella of "envelope-type step-size". Then, we extend the envelope-type step size to the bi-level setting. Finally, we discuss our bi-level line-search algorithms based on Adam and SGD.

**Converging SPSB and SLSB by Envelope Approach** We first propose simple variants of SLS and SPS that converge in the non-interpolating setting while not requiring the step size to be monotonic. To this end, we introduce a new stochastic Polyak step size (SPSB). For comparison, we also recall the step-sizees of $\text{SPS}_{\max}$ and DecSPS . For all methods, the iterate updates are given as $x_{k+1} = x_k - \gamma_k \nabla f_{i_k}(x^k)$ where $i_k$ is sampled uniformly from $[n] = \{1, \dots, n\}$ at each iteration $k$. The step-sizes $\gamma_k$ are then defined as follows:

$$\text{SPS}_{\max} \text{ [29]:} \quad \gamma_k = \min\{\frac{f_{i_k}(x^k) - f_{i_k}^*}{c\|\nabla f_{i_k}(x^k)\|^2}, \gamma_{b,0}\} \tag{5}$$

$$\text{DecSPS [36]:} \quad \gamma_0 = \bar{\gamma} \quad \gamma_k = \frac{1}{c_k}\min\{\frac{f_{i_k}(x^k) - l_{i_k}^*}{\|\nabla f_{i_k}(x^k)\|^2}, c_{k-1}\gamma_{k-1}\} \quad \forall k \geq 1 \tag{6}$$

$$\textbf{SPSB (ours):} \quad \gamma_k = \min\{\frac{f_{i_k}(x^k) - l_{i_k}^*}{c_k\|\nabla f_{i_k}(x^k)\|^2}, \gamma_{b,k}\}, \tag{7}$$

where $f_i^* = \inf_x f_i(x)$, $\bar{\gamma} = \frac{1}{c_0}\min\{\frac{f_{i_0}(x^0) - l_{i_0}^*}{\|\nabla f_{i_0}(x^0)\|^2}, c_0\gamma_{b,0}\}$ , $c_k$ is non-decreasing, $\gamma_{b,k}$ is non-increasing, and $l_i^* \leq f_i^*$ is any lower bound.

Unlike $\text{SPS}_{\max}$ in which $\gamma_{b,0}$ is a constant, our upper bound $\gamma_{b,k}$ is non-increasing. Also, unlike DecSPS in which both the step size and the upper bound are non-increasing (this is because $\gamma_k \leq \frac{c_{k-1}}{c_k}\gamma_{k-1}$ and $\min\{\frac{1}{2cL_{\max}}, \frac{c_0\gamma_{b,0}}{c_k}\} \leq \gamma_k \leq \frac{c_0\gamma_{b,0}}{c_k}$ [36, Lemma 1]), we simplify the recursive structure and do not require the step-size to be monotonic. As we empirically observe in Figure 3, the step size of DecSPS is similar to that of decaying SGD and in fact can be much smaller. Interestingly, the resulting performance of DecSPS is worse than $\text{SPS}_{\max}$ despite $\text{SPS}_{\max}$ eventually becoming unstable once the iterates get closer to the neighborhood of a solution and the step-size naturally behaves erratically. This is not unexpected due to small gradient norms (note the division by gradient-norm in (5)) and dissimilarity between samples in the non-interpolating scenario. Moreover, note that the adaptivity of SPS in the early stage seems to be lost in DecSPS due to the monotonicity of the latter. On the other hand, SPSB not only takes advantage of the large SPS steps that leads to fast convergence, but also stays regularized due to the non-increasing upper bound $\gamma_{b,k}$ in (19). These observations are further supported by the experiments on quadratic functions given in Figure 2, where we observe the fast convergence of SPSB and the instability of $\text{SPS}_{\max}$ . Furthermore, SPSB is significantly more robust to the choice of $\gamma_{b,0}$ than decaying-step SGD as shown by Figure 3a. Motivated by the good practical performance of SPSB , we take a similar approach for SLS. The

SLS proposed and analyzed by Vaswani et al. [44] starts with $\gamma_{b,0}$ and in each iteration $k$ finds the largest $\gamma_k \leq \gamma_{b,0}$ that satisfies:

$$f_{i_k}(x_k - \gamma_k \nabla f_{i_k}(x_k)) \leq f_{i_k}(x_k) - \bar{c} \cdot \gamma_k \|\nabla f_{i_k}(x_k)\|^2, \quad 0 < \bar{c} < 1. \tag{8}$$

To ensure its convergence without interpolation, we replace $\gamma_{b,0}$ with appropriate non-increasing sequence $\gamma_{b,k}$. We name this variant of SLS as SLSB . Interestingly, the empirical performance and step size of SLSB are similar to those of SPSB (see Figure 3). This can be explained by observing that the step sizes of SPSB and SLSB share similar envelope structures, as follows (see Lemma 1 in Appendix A):

$$\text{SPSB}: \quad \min\{\frac{1}{2cL_{\max}}, \gamma_{b,k}\} \leq \gamma_k = \min\{\frac{f_{i_k}(x^k) - l_{i_k}^*}{c\|\nabla f_{i_k}(x^k)\|^2}, \gamma_{b,k}\}, \quad 0 < c,$$

$$\text{SLSB}: \quad \min\{\frac{2(1-\bar{c})}{L_{\max}}, \gamma_{b,k}\} \leq \gamma_k \leq \min\{\frac{f_{i_k}(x^k) - l_{i_k}^*}{\bar{c}\|\nabla f_{i_k}(x^k)\|^2}, \gamma_{b,k}\}, \quad 0 < \bar{c} < 1.$$

Therefore, we unify their analysis based on the following generic *envelope-type step size*:

$$\gamma_k = \min\{\max\{\gamma_{l,k}, \tilde{\gamma}_k\}, \gamma_{b,k}\}, \quad \gamma_{l,k} = \min\{w, \gamma_{b,k}\}, \tag{9}$$

where $\omega > 0$, $\gamma_{b,k}$ is non-increasing, and $\tilde{\gamma}_k$ satisfies $\gamma_{l,k} := \min\{\omega, \gamma_{b,k}\} \leq \tilde{\gamma}_k \leq \gamma_{b,k}$. We show that this envelope-type step size converges at a rate $\mathcal{O}(\frac{1}{\sqrt{K}})$ and $\mathcal{O}(\frac{1}{K})$ for convex and strongly-convex losses respectively.

**Envelope Step Size for Bi-level Optimization (BiSPS)** We extend the use of envelope-type step sizes to the bi-level setting. The step sizes for upper and lower-level objectives of our general envelope-type method are:

Upper: $\alpha_k = \min\{\max\{\alpha_{l,k}, \tilde{\alpha}_k\}, \alpha_{b,k}\}$ hence $\alpha_{l,k} \leq \tilde{\alpha}_k \leq \alpha_{b,k}$ (10)

Lower: $\beta_{k,t} = \min\{\frac{g(x^k, y^{k,t}; \psi) - g(x^k, y^*_{x^k,\psi}; \psi)}{p\|\nabla_y g(x^k, y^{k,t}; \psi)\|^2}, \beta_{b,k}\}$ $\forall t$, (11)

where $y^*_{x^k,\psi}$ is the minimizer of the function $g(x^k, \cdot; \psi)$, and $\alpha_{l,k}$, $\alpha_{b,k}$, and $\beta_{b,k}$ are three non-increasing sequences. Note that $\beta_{b,k}$ is fixed over the lower-level iterations for a given $k$, therefore this is equivalent to running $T$ steps of $\text{SPS}_{\max}$ to minimize the function $g$ at each upper iteration $k$. However, the decrease in the upper bound $\beta_{b,k}$ with $k$ is crucial to guarantee the overall convergence of the algorithm (see Theorem 3). Starting from the general step-size rules in (10) and (11), our bi-level extension of SPS (BiSPS) follows by setting $\alpha_k$ in the form of SPS computed using stochastic hypergradient $h_f^k$. That is,

$$\bar{\alpha}_k = \frac{f(x^k, y^{k+1}; \phi) - l^*_{f(\cdot,y^{k+1};\phi)}}{p\|h_f^k\|^2}, \quad \alpha_{l,k} = \frac{\alpha_{l,0}}{\sqrt{k+1}}, \quad \alpha_{b,k} = \frac{\alpha_{b,0}}{\sqrt{k+1}}, \tag{12}$$

**Algorithm 2** reset

**Input:** $p, q, \eta \geq 1$, opt

**Output:** $p$

1: **if** opt = 1 **then**
2:     $p \leftarrow q$
3: **else if** opt = 2 **then**
4:     $p \leftarrow p$
5: **else if** opt = 3 **then**
6:     $p \leftarrow \eta \cdot p$
7: **end if**

where $\alpha_{l,0} \leq \alpha_{b,0}$ and $l^*_{f(\cdot,y^{k+1};\phi)}$ is a lower bound for $\inf_x f(x, y^{k+1}; \phi)$. For computing $h_f^k$, we can take a similar approach as previous works [15, 19, 4] that use a Neumann series approximation

$$h_f^k = \nabla_x f(x^k, y^{k+1}; \phi) - \nabla_{xy} g(x^k, y^{k+1}; \psi_0) \big[ \frac{N}{L_g} \prod_{j=1}^{\bar{N}} (I - \nabla_{yy}^2 g(x^k, y^{k+1}; \psi_j)) \big] \nabla_y f(x^k, y^{k+1}; \phi),$$

(13)

where $\bar{N}$ is sampled uniformly from $[N]$ and $N$ is the total number of samples. For BiSPS, we use the same sample for $f(x^k, y^{k+1}; \phi)$ and $\nabla_x f(x^k, y^{k+1}; \phi)$ when evaluating $\bar{\alpha}_k$ in (12). Interestingly, we also empirically observe that using independent samples for computing $\bar{\alpha}_k$ and $h_f^k$ results in similar performance as using the same sample. The optimal rate of SGD for non-convex bi-level optimization is $\mathcal{O}(\frac{1}{\sqrt{K}})$ without a growing batch size [4]. We show that BiSPS can obtain the same rate (see Theorem 3) by taking the envelope-type step-size of the form (10) and (11). We implement BiSPS according to (12) and observe that it has better performances over decaying-step SGD with less variations across different values of $\alpha_{b,0}$ (see Figure 4 and note that decaying-step SGD is of the form $\frac{\alpha_{b,0}}{\sqrt{k+1}}$).

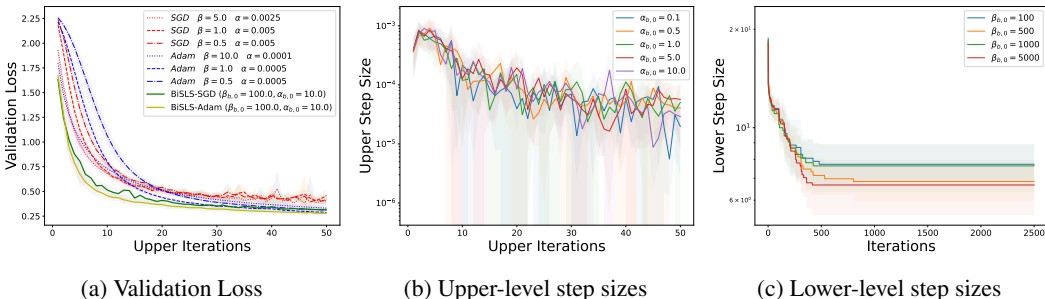

| (a) Validation Loss | (b) Upper-level step sizes | (c) Lower-level step sizes |

Figure 5: Results on hyper-representation learning task (see Sec 4 for details). (a) Validation loss against upper-level iterations for comparing BiSLS-Adam/SGD to fine-tuned Adam/SGD. (b)(c) Upper (left) and lower-level (right) learning rates found by BiSLS-Adam. For the fine-tuned Adam, the optimal lower and upper-level learning rates are $\mathcal{O}(1)$ and $\mathcal{O}(10^{-4})$, respectively. BiSLS-Adam outperforms fine-tuned Adam/SGD with a starting point that is 5 orders of magnitude larger than the optimal step size.

**Stochastic Line-Search Algorithms for Bi-level Optimization**    The challenge of extending SLS to bi-level optimization is rooted in the term $y^*(x)$. In fact, we realize that some of the bi-level objectives are of the form $F(x) = f(y^*(x))$. That is, $f$ does not have an explicit dependence on $x$ as in the data hyper-cleaning task [22]. This implies that when SLS takes a potential step on $x$, the approximation of $y^*(x)$ (i.e, $\bar{y}(x)$) also needs to be updated, otherwise there is no change in function values. Moreover, the use of approximation $\bar{y}(x)$ and the stochastic estimation error in hypergradient would not gaurantee a step size can be always found. To this end, we modify the Armijo line-search rule to be:

BiSLS-SGD:    $f\big(x^k - \alpha_k h_f^k, \hat{y}^{k+1}(x^k - \alpha_k h_f^k)\big) \leq f(x^k, y^{k+1}) - p\alpha_k \|h_f^k\|^2 + \delta,$

BiSLS-Adam:    $f\big(x^k - \alpha_k A_k^{-1} h_f^k, \hat{y}^{k+1}(x^k - \alpha_k A_k^{-1} h_f^k)\big) \leq f(x^k, y^{k+1}) - p\alpha_k \|h_f^k\|_{A_k^{-1}}^2 + \delta,$
$$(14)$$

where $p, \delta > 0$ and $A_k$ is a positive definite matrix such that $A_k^2 = G_k$.

Similar to the single-level Adam case, the matrix $G_k$ in the bi-level setting is defined as $G_k = (\beta_2 G_{k-1} + (1 - \beta_2) \operatorname{diag}(h_f^k h_f^{k^T}))/(1 - \beta_2^k)$ [25, 43]. Moreover, BiSLS-Adam takes the following steps for updating the variable $x$: $x^{k+1} = x^k - \alpha_k A_k^{-1} m_k$ where $m^{k+1} = \beta_1 m^k - (1 - \beta_1)h_f^k$. The details are given in Algorithms 1 and 2. We denote the search starting point for the upper-level as $\alpha_{b,k}$ at iteration $k$, and denote it as $\beta_{b,k}^t$ at step $t$ within iteration $k$ for the lower-level. We remark the following key benefits

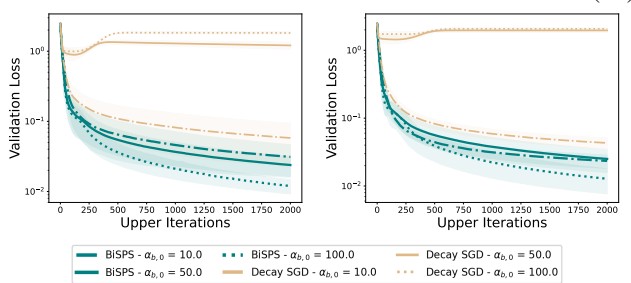

Figure 4: Results on data distillation experiments adapted from Lorraine et al. [30] (see Sec 4 for details). We compare BiSPS and decaying-step SGD for different values of $\alpha_{b,0}$ where Hessian inverse in (2) is computed based on the Identity matrix (left) or Neumann series (right). The lower-level learning rate is fixed at $10^{-4}$.

of resetting $\alpha_{b,k}$ and $\beta_{b,k}^t$ (by using Algorithm 2) to larger values with reference to $\alpha_k$ and $\beta_k^t$ (respectively) at each step: (1) we avoid always searching from $\alpha_{b,0}$ or $\beta_{b,0}^0$, thus reducing computation cost and (2) preserving an overall non-increasing (not necessarily monotonic) trend for $\alpha_{b,k}$ and $\beta_{b,k}^t$ (improving training stability). We found that different values of $\eta$ all give good empirical performances (see Appendix B). The key algorithmic challenge we are facing is that during the backtracking process, for any candidate $\alpha_k$, we need to compute $\hat{x}^k := x^k - \alpha_k h_f^k$ and approximate $y^*(\hat{x}^k)$ with $\hat{y}^{k+1}$ (see Algorithm 1). To limit the cost induced by this nested loop, we limit the number of steps to obtain $\hat{y}^{k+1}$ to be 1.

Moreover, $\delta$ in (14) plays the role of a safeguard that ensures a step size can be found. We set $\delta$ to be small to avoid finding unrealistically large learning rates while tolerating some error in the

hypergradient estimation (see Appendix B for experiments on the sensitivity of $\delta$). In practice, we found that simply setting $\delta = 0$ works well. In Figure 5a, we observe that BiSLS-Adam outperforms fine-tuned Adam or SGD. Surprisingly, its training is stable even when the search starting point $\alpha_{b,0}$ is 5 orders of magnitude larger than a fine-tuned learning rate ($\mathcal{O}(10^{-4})$). Importantly, BiSLS-Adam finds large upper and lower-level learning rates in early phase (see Figure 5b, 5c) for different values of $\alpha_{b,0}$ and $\beta_{b,0}$ that span 3 orders of magnitudes. Interestingly, the learning rates naturally decay with training (also see Figure 6c and 6d). In essence, BiSLS is a **truly adaptive (no knowledge of initialization required) and robust (different initialization works) method that finds large $\alpha$ and $\beta$ without tuning**. In the next section, we give the convergence results of the envelope-type step size.

## 3  Convergence Results

### 3.1  Envelope-type step size for single-level optimization

We first state the assumptions, which are standard in the literature, that will be used for analyzing single-level problems. Assumption 1 is on the Lipschitz continuity of $f$ and $f_i$ in Problem 4.

**Assumption 1.** *The individual function $f_i$ is convex and $L_i$-smooth such that $\|\nabla f_i(x) - \nabla f_i(x')\| \leq L_i\|x - x'\|, \forall i, \forall x \in \mathrm{dom}\, f$ and the overall function $f$ is $L$-smooth. We denote $L_{\max} \triangleq \max_i L_i$. Furthermore, we assume there exists $l_i^*$ such that $l_i^* \leq f_i^* := \inf_x f_i(x), \forall i$, and $f$ is lower bounded by $f^*$ obtained by some $x^*$ such that $f^* = f(x^*)$.*

The following bounded gradient assumption is also used in the analysis of convex problems [41, 33].

**Assumption 2.** *There exists $G > 0$ such that $\|\nabla f_i(x)\|^2 \leq G, \forall i$.*

We first state the theorem for the envelope-type step size defined in (9) for convex functions.

**Theorem 1.** *Suppose Assumption 1, 2 hold, each $f_i$ is convex, $C = \mathrm{dom}\, f$, $\gamma_k$ is independent of the sample $\nabla f_k(x^k)$, and choose $\gamma_{b,k} = \frac{\gamma_{b,0}}{\sqrt{k+1}}$. Then, the envelope-type step size in (9) achieves the following rate,*

$$\mathbb{E}[f(\bar{x}^K) - f(x^*)] \leq \frac{\|x^0 - x^*\|^2}{2\gamma_{l,K-1}K} + \frac{\gamma_{b,0}^2 G^2 \log(K)}{2\gamma_{l,K-1}K},$$

*where $\gamma_{l,K-1} = \min\{\omega, \frac{\gamma_{b,0}}{\sqrt{K}}\}$ and $\bar{x}^K = \frac{1}{K}\sum_{k=0}^{K} x^k$.*

We were not able to give a convergence result that uses the same sample for computing the step size and the gradient. However, we empirically observe that the performance is similar when using either one or two independent samples per iteration (see Figure 2 and Appendix B). When two independent samples $i_k$ and $j_k$ are used per iteration, the first computes the gradient sample $\nabla f_{i_k}(x^k)$, and the other computes the step-size $\gamma_k$. For example, for SPSB this gives $\gamma_k = \min\{\frac{f_{j_k}(x^k) - l_{j_k}^*}{c_k\|\nabla f_{j_k}(x^k)\|^2}, \gamma_{b,k}\}$. This type of assumption has been used in several other works for analyzing adaptive step sizes [27, 44, 29]. Under this assumption, we specialize the results of Theorem 1 to SPSB and SLSB , where $\gamma_{l,K-1} = \min\{\frac{1}{2cL_{\max}}, \frac{\gamma_{b,0}}{\sqrt{K}}\}$ and $\gamma_{l,K-1} = \min\{\frac{2(1-\bar{c})}{L_{\max}}, \frac{\gamma_{b,0}}{\sqrt{K}}\}$ respectively. Concretely, for $K \geq \gamma_{b,0}^2 L_{\max}^2$, SLSB and SPSB with $\gamma_{b,k} = \frac{\gamma_{b,0}}{\sqrt{k+1}}$ and $c = \bar{c} = \frac{1}{2}$ achieve the following rate: $\mathbb{E}[f(\bar{x}^K) - f(x^*)] \leq \frac{\|x^0 - x^*\|^2}{2\gamma_{b,0}\sqrt{K}} + \frac{\gamma_{b,0}G^2 \log(K)}{2\sqrt{K}}$. Next, we state the result for the envelop-type step size when $f$ is $\mu$-strongly convex.

**Theorem 2.** *Suppose a $\mu$-strongly convex function $f$ satisfying Assumptions 1 and 2, assume $\mathcal{C}$ is a closed and convex set, and $\gamma_k$ is independent of the sample $\nabla f_k(x^k)$. Then an envelope-type step size as in (9) with $\gamma_{b,k} = \frac{\gamma_{b,0}}{k+1}$, $\gamma_{b,0} \geq \frac{1}{\mu}$, and $\omega\mu < 1$ achieves the following rate*

$$\mathbb{E}[f(\bar{x}_K) - f(x^*)] \leq \frac{\mu k_0}{2(K - k_0)}\left(e^{-k_0\mu\omega}\|x_0 - x^*\|^2 + \gamma_{b,0}^2 G^2\right) + \frac{\gamma_{b,0}G^2 \log K}{2(K - k_0)},$$

*where $\bar{x}_K = \frac{1}{K - k_0}\sum_{k=k_0}^{K-1} x^k$ and $k_0 = \max\{1, \lceil \gamma_{b,0}/\omega \rceil - 1\}$.*

We can again apply the result of Theorem 2 to SPSB and SLSB with $\gamma_{b,k} = \frac{\gamma_{b,0}}{k+1}$, $\gamma_{b,0} \geq \frac{1}{\mu}$, $\omega = 1/L_{\max}$, and $c = \bar{c} = \frac{1}{2}$ to get an explicit rate: $\mathbb{E}[f(\bar{x}_K) - f(x^*)] \leq \frac{\mu k_0}{2(K-k_0)} \left( e^{\frac{-k_0\mu}{L_{\max}}} \|x_0 - x^*\|^2 + \gamma_{b,0}^2 G^2 \right) + \frac{\gamma_{b,0} G^2 \log K}{2(K-k_0)}$, where $k_0 = \max\{1, \lceil \gamma_{b,0} L_{\max} \rceil - 1\}$.

**Remark 1.** *Under the envelop-type step size framework and the assumption of two independent samples, SLSB and SPSB share the same convergence rates of $\mathcal{O}(\frac{1}{\sqrt{K}})$ and $\mathcal{O}(\frac{1}{K})$ as SGD with decaying step-size for convex and strongly-convex losses respectively. For the latter, a projection operation is required to stay in the closed and convex set $\mathcal{C}$. These rates are not surprising because of the structure of the envelope step-size in* (9). *Indeed, the proof is similar to the standard proof of analogous rate for SGD with decaying step-size. Nonetheless, we include it here for completeness.*

### 3.2 Envelope-type step size for bi-level optimization

We start with recalling standard assumptions in BO [22, 19, 15, 4]. We denote $z = [x; y]$ and recall the bi-level problem in (1). The first assumption is on the lower-level objective $g$.

**Assumption 3.** *The function $g(x, y)$ is $\mu_g$ strongly convex in $y$ for any given $x$. Moreover, $\nabla g$ is Lipschitz continuous: $\|\nabla g(x_1, y_1) - \nabla g(x_2, y_2)\| \leq L_g \|z_1 - z_2\|$ (also assume that this holds true for each sampled function $g(x, y; \psi)$), and $\nabla^2 g$ is Lipschitz continuous: $\|\nabla^2 g(x_1, y_1) - \nabla^2 g(x_2, y_2)\| \leq L_G \|z_1 - z_2\|$. We further assume that $\|\nabla_{xy}^2 g(x, y)\| \leq C_g$, and the condition number is defined as $\kappa = \frac{L_g}{\mu_g}$.*

Next, we state the assumptions on the upper objective $f$.

**Assumption 4.** *The function $f$ and its gradients are Lipschitz continuous. That is: $\|f(x_1, y_1) - f(x_2, y_2)\| \leq L_1 \|z_1 - z_2\|$ and $\|\nabla f(x_1, y_1) - \nabla f(x_2, y_2)\| \leq L_{f,1} \|z_1 - z_2\|$. We also assume that $\|\nabla_y f(x, y)\| \leq C_f$.*

Furthermore, we make the following standard assumptions on the estimates of $\nabla f$, $\nabla g$, and $\nabla^2 g$.

**Assumption 5.** *The stochastic gradients are unbiased: $\mathbb{E}_\phi[\nabla f(x, y; \phi)] = \nabla f(x, y)$, $\mathbb{E}_\psi[\nabla g(x, y; \psi)] = \nabla g(x, y)$, and $\mathbb{E}_\psi[\nabla^2 g(x, y; \psi)] = \nabla^2 g(x, y)$. The variances of $\nabla f(x, y; \phi)$ and $\nabla^2 g(x, y; \psi)$) are bounded: $\mathbb{E}_\phi[\|\nabla f(x, y; \phi) - \nabla f(x, y)\|^2] \leq \sigma_f^2$ and $\mathbb{E}_\psi[\|\nabla^2 g(x, y; \psi) - \nabla^2 g(x, y)\|^2] \leq \sigma_G^2$.*

Finally, we introduce the bounded optimal function value assumption in (15), which is used specifically for analyzing step size of the form (11) in the bi-level setting:

$$\mathbb{E}_\psi[g(x, y^*(x); \psi) - g(x, y_{x,\psi}^*; \psi)] \leq \sigma_g^2, \forall x, \tag{15}$$

$$\mathbb{E}[\|\nabla_y g(x, y) - \nabla_y g(x, y; \phi)\|^2] \leq \sigma_g^2, \forall x, y, \tag{16}$$

where $y^*(x) = \min_y g(x, y)$ and $y_{x,\psi}^* = \min_y g(x, y; \psi)$ for a given $x$ (recall that at any iteration $k$, the lower-level steps in BiSPS are $\text{SPS}_{\max}$ with an upper bound $\beta_{b,k}$; furthermore, $\beta_{b,k}$ is non-increasing w.r.t. upper iteration $k$). The one-variable analogous assumption of (15) has been used in the analysis of $\text{SPS}_{\max}$ [29]. Here, we extend it to a two-variable function. Unlike the bounded variance assumption (16), which needs to hold true for all $x$ and $y$, we require (15) to hold at $y^*(x)$ for any given $x$. As mentioned previously, the closed form solution $y^*(x)$ is difficult to obtain. Thus, we define the following expression by replacing $y^*(x)$ with $y$ in (2):

$$\bar{\nabla} f(x, y) = \nabla_x f(x, y) - \nabla_{xy}^2 g(x, y) [\nabla_{yy}^2 g(x, y)]^{-1} \nabla_y f(x, y). \tag{17}$$

A stochastic Neumann series in (13) approximates (17) with $x$ and $y$ being $x^k$ and $y^{k+1}$ (respectively), also recall that $y^{k+1}$ is an approximation of $y^*(x^k)$ by running $T$ lower-level SGD steps to minimize $g$ w.r.t. $y$ for a fixed $x^k$. Based on Assumptions 3, 4, and 5, we have the following results to be used in the analysis [15]: $\|\nabla F(x_1) - \nabla F(x_2)\| \leq L_F \|x_1 - x_2\|$, $\|y^*(x_1) - y^*(x_2)\| \leq L_y \|x_1 - x_2\|$, and $\|\bar{\nabla} f(x, y^*(x)) - \bar{\nabla} f(x, y)\| \leq L_f \|y^*(x) - y\|$. Furthermore, the bias in the stochastic hypergradient in (13) (denoted as $B$) decays exponentially with $N$ and its variance is bounded, i.e. $\mathbb{E}[\|h_f^k - \mathbb{E}[h_f^k]\|^2] \leq \tilde{\sigma}_f^2$ (see Appendix A for details) [19].

Now, we state our main theorem based on step size of the form (10) and (11).

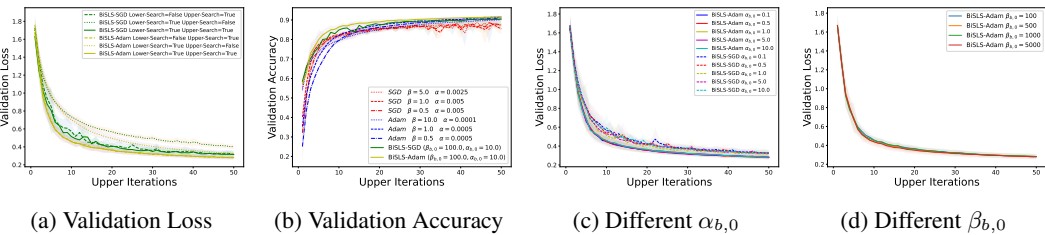

| (a) Validation Loss | (b) Validation Accuracy | (c) Different $\alpha_{b,0}$ | (d) Different $\beta_{b,0}$ |

Figure 6: Validation loss (a) and accuracy (b) against iterations. (a) Comparisons between whether to use or not use line-search at the upper or lower level; (b) Generalization performance of BiSLS-Adam/SGD and fine-tuned Adam/SGD; (c) Validation loss against iterations for different values of $\alpha_{b,0}$ ($\beta_{b,0}$ fixed at 100). (d) Same plot as (c) but for different values of $\beta_{b,0}$ ($\alpha_{b,0}$ fixed at 10).

**Theorem 3.** *Suppose $f$ and $g$ satisfy Assumptions 3, 4, and 5, learning-rate upper bounds $\alpha_{b,k} = \frac{\alpha_{b,0}}{\sqrt{k+1}}$ and $\alpha_{l,k} = \frac{\alpha_{l,0}}{\sqrt{k+1}}$ with $\alpha_{b,0}$ and $\alpha_{l,0}$ satisfying $\frac{1}{L_F + 4L_y^2} \geq \frac{\alpha_{b,0}^2}{\alpha_{l,0}}$ and $\alpha_{l,0} \leq \alpha_{b,0}$. Further assume that $\alpha_k$ is independent of the stochastic hypergradient $h_f^k$, and each sampled function $g(x, y; \psi)$ is convex. Then under the Assumption (15) with $p \geq \frac{1}{2}$, $C_k = \min\{\frac{1}{2pL_g}, \beta_{b,k}\}$, $T \geq \frac{\log(\alpha_{b,0} L_f^2 + 2)}{-\log(1 - \mu_g C_{K-1})}$, and $\beta_{b,k} = \frac{\beta_{b,0}}{k+1}$, BiSPS achieves the rate:*

$$\frac{1}{K} \sum_{k=0}^{K-1} \mathbb{E}[\|\nabla F(x^k)\|^2] \leq \tilde{\mathcal{O}}(\frac{\kappa^3}{\sqrt{K}} + \frac{\kappa^2 \log K}{\sqrt{K}}). \tag{18}$$

**Remark 2.** *We further give the convergence result under the bounded variance assumption (16) in Appendix A. Theorem 3 shows that BiSPS matches the optimal rate of SGD up to a logarithmic factor without a growing batch size. We notice that the assumption (15) largely simplifies the expression on $T$ and does not require an explicit upper bound on $\beta_{b,0}$. As in the single-level case, whether using one sample or two samples (which makes upper-level step-size independent of gradient) gives similar empirical performances (see Appendix B). Note that the independence assumption is only needed for the upper-level. Thus, the two-sample requirement of theorem does not apply to the lower-level problem. This is useful from computational standpoint as typical bi-level algorithms run multiple lower-level updates for each upper-level iteration.*

## 4 Additional Hyper-Representation and Data Distillation Experiments

**Hyper-representation learning:** The experiments are performed on MNIST dataset using LeNet [26, 42]. We use conjugate gradient method for solving system of equations when computing the hypergradient [17]. The upper and lower-level objectives are to optimize the embedding layers and the classifier (i.e. the last layer of the neural net), respectively (see Appendix B for details). For constant-step SGD and Adam, we tune the lower-level learning rate $\beta \in \{10.0, 5.0, 1.0, 0.5, 0.1, 0.05, 0.01\}$. For the upper-level learning rate, we tune $\alpha \in \{0.001, 0.0025, 0.005, 0.01, 0.05, 0.1\}$ for SGD, and $\alpha \in \{10^{-5}, 5 \cdot 10^{-5}, 10^{-4}, 5 \cdot 10^{-4}, 0.001, 0.01\}$ for Adam (recall that $\delta$ in (14) is set to 0). Based on the results of Figure 6, we make the following key observations: ① **line-search at the upper-level is essential for achieving the optimal performance (Figure 6a); ② BiSLS-Adam/SGD not only converges fast but also generalizes well (Figure 6b); ③ BiSLS-Adam/SGD is highly robust to search starting points $\alpha_{b,0}$ and $\beta_{b,0}$ (Figure 6c, 6d). It addresses the fundamental question of how to tune $\alpha$ and $\beta$ in bi-level optimization** (see Appendix B for additional results on search cost).

**Search cost and run-time comparison:** The search cost of reset option 1 in Algorithm 2 for BiSLS-SGD and BiSLS-Adam is $89 \pm 15$ and $115 \pm 16$ per iteration, respectively, measured in the number of condition-checking via (14). These costs are significantly reduced when option 3 is used. Concretely, Figure 7b suggests that choosing $\eta = 2$ in option 3 results in $\sim 9$ evaluations of (14) per iteration for both BiSLS-Adam and BiSLS-SGD. Choosing $\eta = 1$ in reset option 3 is equivalent to reset option 2, which further cuts down the cost to $\sim 4$. However, this choice forces the learning rate to be monotonically-decreasing, which leads to a slower convergence compared to other $\eta$'s (see Appendix B for more details). Besides this, Figure 7a shows that a single step for

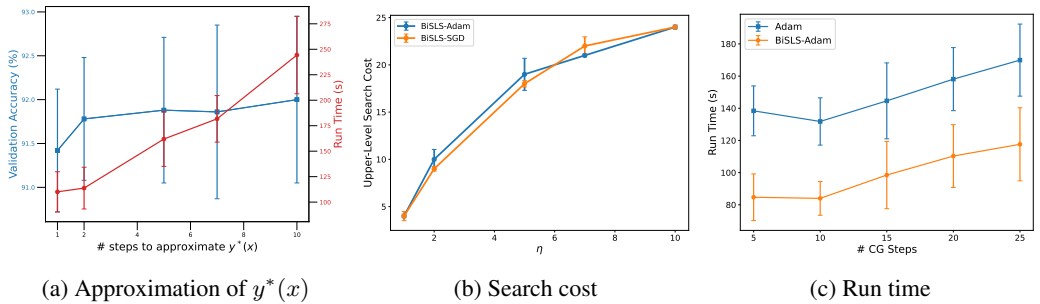

| (a) Approximation of $y^*(x)$ | (b) Search cost | (c) Run time |

Figure 7: (a) Validation accuracy and run time (in seconds) against different number of gradient steps for approximating $y^*(x)$ in (14). (b) Search cost (i.e. number of evaluations of (14) per iteration) against different $\eta$'s for reset option 3 in Algorithm 2. (c) Run time (in seconds) of BiSLS-Adam and fine-tuned Adam to reach $85\%$ validation accuracy against different number of conjugate gradient steps for computing the hypergradient.

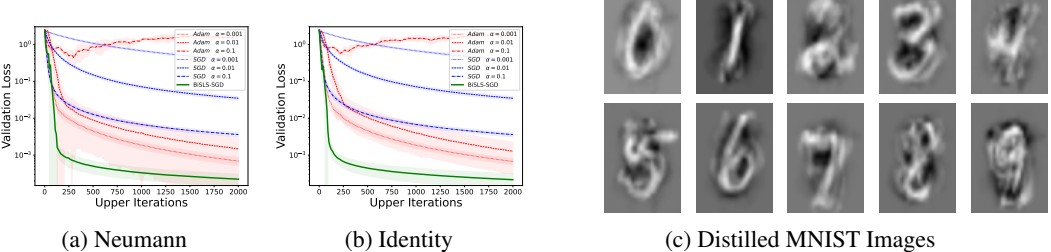

| (a) Neumann | (b) Identity | (c) Distilled MNIST Images |

Figure 8: (a)(b): Comparison between BiSLS-SGD and Adam/SGD for Data Distillation on MNIST dataset. Validation loss plotted against iterations. (a) Hypergradient computed using Neumann series; (b) Inverse Hessian in (2) treated as the Identity [30] when computing the hypergradient; (c) Distilled MNIST images after 3000 iterations of BiSLS-SGD.

approximating the $y^*(x)$ in (14) is sufficient, considering the trade-off between performance gain and extra computation overhead introduced. Most importantly, BiSPS-Adam takes less time to reach a threshold test accuracy than fine-tuned Adam (Figure 7c) due to a suitable (and potentially large) learning rate found using (14) (note that Figure 7c excludes the cost of tuning the learning rate of Adam).

**Data distillation:** The goal of data distillation is to generate a small set of synthetic data from an original dataset that preserves the performance of a model when trained using the generated data [46, 50]. We adapted the experiment set up from Lorraine et al. [30] to distill MNIST digits. We present the results in Figure 8, where we observe that BiSLS-SGD converges significantly faster than fine-tuned Adam or SGD, and generate realistic MNIST images (see Appendix B for more results).

## 5 Conclusion

In this work, we have given simple alternatives to SLS and SPS that show good empirical performance in non-interpolating scenario without requiring the step size to be monotonic. We unify their analysis based on a simplified envelope-type step size, and extend the analysis to the bi-level setting while designing a SPS-based bi-level algorithm. In the end, we propose bi-level line-search algorithm BiSLS-Adam/SGD that is empirically truly robust and adaptive to learning rate initialization. Our work opens several possible future directions. Given the superior performance of BiSLS, we prioritize an analysis of its convergence rates. The difficulty stems from: (a) the bias in hypergradient estimation; (b) the dual updates in $x$ and $y^*(x)$ (incurring nested loop structures); (c) the error in estimating $y^*(x)$. On single-level optimization, we remark as an important direction to relax the two-sample assumption on SPSB /SLSB . Ultimately, we hope to promote further research on bi-level optimization algorithms with minimal tuning.

**Acknowledgement** This work is funded partially by the NSERC Discovery Grant RGPIN-2021-03677, the NSERC Discovery Grant RGPIN-2022-03669, and the Canada CIFAR AI Chair Program.

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

# A  Proofs of Theorems and Additional Convergence Results

## A.1  Useful Lemmas

Lemma 1 provides more details on the envelope structure of SPSB and SLSB given in (10) and (11). The lower bound in (19) will also be used in Lemma 5 (for bounding the term $\|y^{k+1} - y^*(x^k)\|^2$).

**Lemma 1.** *Under the Assumption 1, we have the following:*

$$SPSB: \quad \min\{\frac{1}{2cL_{\max}}, \gamma_{b,k}\} \leq \gamma_k = \min\{\frac{f_{i_k}(x^k) - l_{i_k}^*}{c\|\nabla f_{i_k}(x^k)\|^2}, \gamma_{b,k}\}, \quad 0 < c, \tag{19}$$

$$SLSB: \quad \min\{\frac{2(1-\bar{c})}{L_{\max}}, \gamma_{b,k}\} \leq \gamma_k \leq \min\{\frac{f_{i_k}(x^k) - l_{i_k}^*}{\bar{c}\|\nabla f_{i_k}(x^k)\|^2}, \gamma_{b,k}\}, \quad 0 < \bar{c} < 1. \tag{20}$$

*Proof.* The bounds in (19) have been shown in [29, 36]. For (20), the first part of the inequality has been shown in [44]. For the second part, recall the Armijo condition (14):

$$f_{i_k}(x_k - \gamma_k \nabla f_{i_k}(x_k)) \leq f_{i_k}(x_k) - \bar{c} \cdot \gamma_k \|\nabla f_{i_k}(x_k)\|^2, \quad 0 < \bar{c} < 1.$$

We can then rearrange this to obtain

$$\gamma_k \leq \frac{f_{i_k}(x_k) - f_{i_k}(x_k - \gamma_k \nabla f_{i_k}(x_k))}{\bar{c}\|\nabla f_{i_k}(x^k)\|^2} \leq \frac{f_{i_k}(x_k) - f_{i_k}^*}{\bar{c}\|\nabla f_{i_k}(x^k)\|^2} \leq \frac{f_{i_k}(x_k) - l_{i_k}^*}{\bar{c}\|\nabla f_{i_k}(x^k)\|^2}, \tag{21}$$

where $l_{i_k}^*$ is any lower bound for $f_{i_k}^*$. Also recall that $\gamma_{b,k}$ is the search starting point at iteration $k$, hence (20) holds for SLSB .  □

Lemma 2 gives the expressions for the constants $L_F$, $L_y$, and $L_f$. The proof can be found in [15, Lem 2.2].

**Lemma 2.** *Under Assumptions 3 and 4, we have the following:*

$$\|\nabla F(x_1) - \nabla F(x_2)\| \leq L_F \|x_1 - x_2\|,$$
$$\|y^*(x_1) - y^*(x_2)\| \leq L_y \|x_1 - x_2\|,$$
$$\|\bar{\nabla} f(x, y^*(x)) - \bar{\nabla} f(x, y)\| \leq L_f \|y^*(x) - y\|,$$

*where*

$$L_f = L_{f,1} + \frac{L_{f,1} L_g}{\mu_g} + \frac{L_1}{\mu_g}(L_G + \frac{L_g L_G}{\mu_g}) \sim \mathcal{O}(\kappa^2)$$

$$L_y = \frac{L_g}{\mu_g} \sim \mathcal{O}(\kappa)$$

$$L_F = L_{f,1} + \frac{L_g(L_{f,1} + L_f)}{\mu_g} + \frac{L_1}{\mu_g}(L_G + \frac{L_g L_G}{\mu_g}) \sim \mathcal{O}(\kappa^3)$$

Lemma 3 is on the bias and variance of the stochastic hypergradient in (13), which has the following form [19]

$$h_f^k = \nabla_x f(x^k, y^{k+1}; \phi) - \nabla_{xy} g(x^k, y^{k+1}; \psi_0) \big[\frac{N}{L_g} \prod_{j=1}^{\bar{N}} (I - \nabla_{yy}^2 g(x^k, y^{k+1}; \psi_j))\big] \nabla_y f(x^k, y^{k+1}; \phi). \tag{22}$$

Recall that the hypergradient surrogate defined in (17) based on $(x^k, y^{k+1})$ is

$$\bar{\nabla} f(x^k, y^{k+1}) = \nabla_x f(x^k, y^{k+1}) + \nabla_{xy}^2 g(x^k, y^{k+1})[\nabla_{yy}^2 g(x^k, y^{k+1})]^{-1} \nabla_y f(x^k, y^{k+1}). \tag{23}$$

Given a filtration $\mathcal{F}_k'$ up to and including $x^k$ and $y^{k+1}$, the bias of the stochastic hypergradient is defined as $B = \bar{\nabla} f(x^k, y^{k+1}) - \mathbb{E}[h_f^k | \mathcal{F}_k']$, and the variance is defined as $\mathbb{E}[\|\bar{\nabla} f(x^k, y^{k+1}) - \mathbb{E}[h_f^k | \mathcal{F}_k']\|^2]$. Lemma 3 has been proven in [19, Lem 1.] (also see [4, Lem 5.]). Lemmas 1, 2, and 3 will be used in the proofs of Theorems 3 and 4.

**Lemma 3.** *Under Assumptions 3, 4, and 5, the bias and variance of the stochastic hypergradient $h_f^k$ satisfy the following*

$$\text{Bias: } \|\bar{\nabla} f(x^k, y^{k+1}) - \mathbb{E}[h_f^k|\mathcal{F}_k']\| \le \frac{C_g C_f}{\mu_g}(1 - \frac{\mu_g}{L_g})^N, \forall k$$

$$\text{Variance: } \mathbb{E}[\|\bar{\nabla} f(x^k, y^{k+1}) - \mathbb{E}[h_f^k|\mathcal{F}_k']\|^2] \le \tilde{\sigma}_f^2, \forall k,$$

*where N is the total number of samples, and $\tilde{\sigma}_f^2 = \sigma_f^2 + [(\sigma_f^2 + L_1^2)(\sigma_G^2 + 2L_g^2) + \sigma_f^2 L_g^2]\frac{3}{\mu_g^2} \sim \mathcal{O}(\kappa^2)$.*

Lemma 4 is on the descent of the quantity $V^k := F(x^k) - F(x^*) + \|y^k - y^*(x^k)\|^2$. It will be used in the proofs of Theorem 3 and 4.

**Lemma 4.** *Suppose F satisfies Assumptions 3, 4, and 5, sequences $\alpha_{b,k} = \frac{\alpha_{b,0}}{\sqrt{k+1}}$ and $\alpha_{l,k} = \frac{\alpha_{l,0}}{\sqrt{k+1}}$ with $\alpha_{b,0}$ and $\alpha_{l,0}$ satisfying $\frac{1}{L_F + 4L_y^2} \ge \frac{\alpha_{b,0}^2}{\alpha_{l,0}}$ and $\alpha_{l,0} \le \alpha_{b,0}$. Further assume that $\alpha_k$ is independent of $h_f^k$. Then step sizes of the form (10) and (11) achieve the following:*

$$E[V^{k+1}] \le \mathbb{E}[V^k] - \frac{\alpha_{l,k}}{2}\mathbb{E}[\|\nabla F(x^k)\|^2] + (\alpha_{b,k}L_f^2 + 2)\mathbb{E}[\|y^{k+1} - y^*(x^k)\|^2] +$$

$$\alpha_{b,k}B^2 + (2L_y^2\alpha_{b,k}^2 + \frac{L_F\alpha_{b,k}^2}{2})\tilde{\sigma}_f^2 - \mathbb{E}[\|y^k - y^*(x^k)\|^2], \forall k, \tag{24}$$

*where $V^k = F(x^k) - F(x^*) + \|y^{k+1} - y^*(x^{k+1})\|^2$ and recall that F is the upper-level loss defined in (1).*

*Proof.* We denote $\mathbb{E}[h_f^k|\mathcal{F}_k'] = \bar{h}_f^k$. By the $L_F$-smoothness of the objective $F$:

$$F(x^{k+1}) \le F(x^k) + \langle \nabla F(x^k), x^{k+1} - x^k \rangle + \frac{L_F}{2}\|x^{k+1} - x^k\|^2.$$

Take expectation conditioned on a filtration of past iterates $\mathcal{F}_k'$ (up to and include $x^k, y^{k+1}$):

$$\mathbb{E}[F(x^{k+1})|\mathcal{F}_k'] \le F(x^k) + \mathbb{E}[\langle \nabla F(x^k), x^{k+1} - x^k \rangle|\mathcal{F}_k'] + \frac{L_F}{2}\mathbb{E}[\|x^{k+1} - x^k\|^2|\mathcal{F}_k']$$

$$= F(x^k) - \mathbb{E}[\alpha_k \langle \nabla F(x^k), h_f^k \rangle|\mathcal{F}_k'] + \frac{L_F\mathbb{E}[\alpha_k^2|\mathcal{F}_k']}{2}\mathbb{E}[\|h_f^k - \bar{h}_f^k + \bar{h}_f^k\|^2|\mathcal{F}_k']$$

$$\overset{(a)}{=} F(x^k) - \mathbb{E}[\alpha_k|\mathcal{F}_k']\langle \nabla F(x^k), \bar{h}_f^k \rangle + \frac{L_F\mathbb{E}[\alpha_k^2|\mathcal{F}_k']}{2}\|\bar{h}_f^k\|^2 + \frac{L_F\mathbb{E}[\alpha_k^2|\mathcal{F}_k']}{2}\tilde{\sigma}_f^2$$

$$= F(x^k) - \frac{\mathbb{E}[\alpha_k|\mathcal{F}_k']}{2}\|\nabla F(x^k)\|^2 - \frac{\mathbb{E}[\alpha_k|\mathcal{F}_k']}{2}\|\bar{h}_f^k\|^2 + \frac{\mathbb{E}[\alpha_k|\mathcal{F}_k']}{2}\|\nabla F(x^k) - \bar{h}_f^k\|^2 +$$

$$\frac{L_F\mathbb{E}[\alpha_k^2|\mathcal{F}_k']}{2}\|\bar{h}_f^k\|^2 + \frac{L_F\mathbb{E}[\alpha_k^2|\mathcal{F}_k']}{2}\tilde{\sigma}_f^2,$$

where (a) is by the assumption that $\alpha_k$ is independent of $h_f^k$. Then expand the term $\|\nabla F(x^k) - \bar{h}_f^k\|^2$ as follows:

$$\|\nabla F(x^k) - \bar{h}_f^k\|^2 = \|\nabla F(x^k) - \bar{\nabla} f(x^k, y^{k+1}) + \bar{\nabla} f(x^k, y^{k+1}) - \bar{h}_f^k\|^2$$

$$\le 2\|\nabla F(x^k) - \bar{\nabla} f(x^k, y^{k+1})\|^2 + 2\|\bar{\nabla} f(x^k, y^{k+1}) - \bar{h}_f^k\|^2$$

$$\overset{(b)}{\le} 2L_f^2\|y^{k+1} - y^*(x^k)\|^2 + 2B^2,$$

where (b) is by Lemma 2 and 3. Substituting this into the above:

$$\mathbb{E}[F(x^{k+1})|F_k'] \le F(x^k) - \frac{\mathbb{E}[\alpha_k|F_k']}{2}\|\nabla F(x^k)\|^2 - \frac{\mathbb{E}[\alpha_k|F_k']}{2}\|\bar{h}_f^k\|^2 + \frac{L_F\mathbb{E}[\alpha_k^2|F_k']}{2}\|\bar{h}_f^k\|^2 +$$

$$E[\alpha_k|\mathcal{F}_k']L_f^2\|y^{k+1} - y^*(x^k)\|^2 + E[\alpha_k|\mathcal{F}_k']B^2 + \frac{L_F\mathbb{E}[\alpha_k^2|\mathcal{F}_k']}{2}\tilde{\sigma}_f^2$$

$$\overset{(c)}{\le} F(x^k) - \frac{\alpha_{l,k}}{2}\|\nabla F(x^k)\|^2 - \frac{\alpha_{l,k}}{2}\|\bar{h}_f^k\|^2 + \frac{L_F\alpha_{b,k}^2}{2}\|\bar{h}_f^k\|^2 +$$

$$\alpha_{b,k}L_f^2\|y^{k+1} - y^*(x^k)\|^2 + \alpha_{b,k}B^2 + \frac{L_F\alpha_{b,k}^2}{2}\tilde{\sigma}_f^2,$$

where (c) is by $\alpha_{l,k} \leq \alpha_k$ and $\alpha_{b,k} \geq \alpha_k$. Then take total expectations and subtract $F(x^*)$:

$$\mathbb{E}[F(x^{k+1}) - F(x^*)] \leq \mathbb{E}[F(x^k) - F(x^*)] - \frac{\alpha_{l,k}}{2}\mathbb{E}[\|\nabla F(x^k)\|^2] - \frac{\alpha_{l,k}}{2}\mathbb{E}[\|\bar{h}_f^k\|^2] + \frac{L_F \alpha_{b,k}^2}{2}\mathbb{E}[\|\bar{h}_f^k\|^2] +$$

$$\alpha_{b,k}L_f^2\mathbb{E}[\|y^{k+1} - y^*(x^k)\|^2] + \alpha_{b,k}B^2 + \frac{L_F \alpha_{b,k}^2}{2}\tilde{\sigma}_f^2 \qquad (25)$$

Now define the potential function $V^k := F(x^k) - F(x^*) + \|y^{k+1} - y^*(x^{k+1})\|^2$ and expand the term $\|y^{k+1} - y^*(x^{k+1})\|^2$ as follows:

$$\begin{aligned}
\|y^{k+1} - y^*(x^{k+1})\|^2 &= \|y^{k+1} - y^*(x^k) + y^*(x^k) - y^*(x^{k+1})\|^2 \\
&\leq 2\|y^{k+1} - y^*(x^k)\|^2 + 2\|y^*(x^k) - y^*(x^{k+1})\|^2 \\
&\overset{(d)}{\leq} 2\|y^{k+1} - y^*(x^k)\|^2 + 2L_y^2\|x^{k+1} - x^k\|^2 \\
&= 2\|y^{k+1} - y^*(x^k)\|^2 + 2L_y^2\alpha_k^2\|h_f^k\|^2 \\
&= 2\|y^{k+1} - y^*(x^k)\|^2 + 2L_y^2\alpha_k^2\|h_f^k - \bar{h}_f^k + \bar{h}_f^k\|^2,
\end{aligned}$$

where (d) is by Lemma 2. Take expectation conditioned on $\mathcal{F}_k'$:

$$\begin{aligned}
\mathbb{E}[\|y^{k+1} - y^*(x^{k+1})\|^2 | \mathcal{F}_k'] &\leq 2\|y^{k+1} - y^*(x^k)\|^2 + 2L_y^2\alpha_k^2\|\bar{h}_f^k\|^2 + 2L_y^2\alpha_k^2\tilde{\sigma}_f^2 \\
&\leq 2\|y^{k+1} - y^*(x^k)\|^2 + 2L_y^2\alpha_{b,k}^2\|\bar{h}_f^k\|^2 + 2L_y^2\alpha_{b,k}^2\tilde{\sigma}_f^2.
\end{aligned}$$

Then, take total expectations:

$$\mathbb{E}[\|y^{k+1} - y^*(x^{k+1})\|^2] \leq 2\mathbb{E}[\|y^{k+1} - y^*(x^k)\|^2] + 2L_y^2\alpha_{b,k}^2\mathbb{E}[\|\bar{h}_f^k\|^2] + 2L_y^2\alpha_{b,k}^2\tilde{\sigma}_f^2. \qquad (26)$$

Now, based on the definition of $V^k$ and combining (25) and (26):

$$\begin{aligned}
E[V^{k+1}] &\leq \mathbb{E}[F(x^k) - F(x^*)] - \frac{\alpha_{l,k}}{2}\mathbb{E}[\|\nabla F(x^k)\|^2] \\
&\quad - \frac{\mathbb{E}[\|\bar{h}_f^k\|^2]}{2}(\alpha_{l,k} - L_F\alpha_{b,k}^2 - 4L_y^2\alpha_{b,k}^2) + (\alpha_{b,k}L_f^2 + 2)\mathbb{E}[\|y^{k+1} - y^*(x^k)\|^2] + \\
&\quad \alpha_{b,k}B^2 + (2L_y^2\alpha_{b,k}^2 + \frac{L_F\alpha_{b,k}^2}{2})\tilde{\sigma}_f^2 \\
&\overset{(e)}{\leq} \mathbb{E}[F(x^k) - F(x^*)] - \frac{\alpha_{l,k}}{2}\mathbb{E}[\|\nabla F(x^k)\|^2] + (\alpha_{b,k}L_f^2 + 2)\mathbb{E}[\|y^{k+1} - y^*(x^k)\|^2] + \\
&\quad \alpha_{b,k}B^2 + (2L_y^2\alpha_{b,k}^2 + \frac{L_F\alpha_{b,k}^2}{2})\tilde{\sigma}_f^2 \\
&= \mathbb{E}[V^k] - \frac{\alpha_{l,k}}{2}\mathbb{E}[\|\nabla F(x^k)\|^2] + (\alpha_{b,k}L_f^2 + 2)\mathbb{E}[\|y^{k+1} - y^*(x^k)\|^2] + \\
&\quad \alpha_{b,k}B^2 + (2L_y^2\alpha_{b,k}^2 + \frac{L_F\alpha_{b,k}^2}{2})\tilde{\sigma}_f^2 - \mathbb{E}[\|y^k - y^*(x^k)\|^2],
\end{aligned}$$

where (e) is because $\frac{1}{L_F + 4L_y^2} \geq \frac{\alpha_{b,0}^2}{\alpha_{l,0}}$, which guarantees that $\alpha_{l,k} = \frac{\alpha_{l,0}}{\sqrt{k+1}} \geq (L_F + 4L_y^2)\alpha_{b,k} = (L_F + 4L_y^2)\frac{\alpha_{b,0}^2}{k+1}$. $\qquad \square$

Lemma 5 and Lemma 6 give two alternatives for bounding the term $\|y^{k+1} - y^*(x^k)\|^2$. Lemma 5 is based on the assumption $\mathbb{E}_\psi[g(x, y^*(x); \psi) - g(x, y_{x,\psi}^*; \psi)] \leq \sigma_g^2, \forall x$. The proof for its one-variable analogous assumption is given in Loizou et al. [29]. Here we follow a similar approach for the two-variable function $g(x, y)$. Lemma 5 will be used in the proof of Theorem 3.

**Lemma 5.** *Suppose Assumptions 3, 5, and the bounded optimal function value assumption* (15) *hold. Further assume that each sampled function $g(x, y; \psi)$ is convex, then step size of the form 11 achieves the following:*

$$\mathbb{E}[\|y^{k+1} - y^*(x^k)\|^2] \leq (1 - \mu_g C_k)^T \mathbb{E}[\|y^k - y^*(x^k)\|^2] + 2\beta_{b,k}T\sigma_g^2,$$

*where $C_k = \min\{\frac{1}{2pL_g}, \beta_{b,k}\}$.*

*Proof.* We denote $h_g^{k,t} = \nabla_y g(x^k, y^{k,t}; \psi)$, and $\mathcal{F}'_{k,t}$ be a filtration up to and including $x^k$ and $y^{k,t}$. We have,

$$
\begin{aligned}
\|y^{k,t+1} - y^*(x^k)\|^2 &= \|y^{k,t} - \beta_{k,t} h_g^{k,t} - y^*(x^k)\|^2 \\
&= \|y^{k,t} - y^*(x^k)\|^2 - 2\beta_{k,t}\langle y^{k,t} - y^*(x^k), h_g^{k,t}\rangle + \beta_{k,t}^2 \|h_g^{k,t}\|^2 \\
&\overset{(a)}{\leq} \|y^{k,t} - y^*(x^k)\|^2 - 2\beta_{k,t}\langle y^{k,t} - y^*(x^k), h_g^{k,t}\rangle + \frac{\beta_{k,t}}{p}[g(x^k, y^{k,t}; \psi) - g(x^k, y^*_{x^k,\psi}; \psi)] \\
&\overset{(b)}{\leq} \|y^{k,t} - y^*(x^k)\|^2 - 2\beta_{k,t}\langle y^{k,t} - y^*(x^k), h_g^{k,t}\rangle + 2\beta_{k,t}[g(x^k, y^{k,t}; \psi) - g(x^k, y^*_{x^k,\psi}; \psi)] \\
&= \|y^{k,t} - y^*(x^k)\|^2 - 2\beta_{k,t}\langle y^{k,t} - y^*(x^k), h_g^{k,t}\rangle + \\
&\quad 2\beta_{k,t}[g(x^k, y^{k,t}; \psi) - g(x^k, y^*(x^k); \psi) + g(x^k, y^*(x^k); \psi) - g(x^k, y^*_{x^k,\psi}; \psi)] \\
&= \|y^{k,t} - y^*(x^k)\|^2 + 2\beta_{k,t}[-\langle y^{k,t} - y^*(x^k), h_g^{k,t}\rangle + g(x^k, y^{k,t}; \psi) - g(x^k, y^*(x^k); \psi)] + \\
&\quad 2\beta_{k,t}[g(x^k, y^*(x^k); \psi) - g(x^k, y^*_{x^k,\psi}; \psi)] \\
&\overset{(c)}{\leq} \|y^{k,t} - y^*(x^k)\|^2 + 2C_k[-\langle y^{k,t} - y^*(x^k), h_g^{k,t}\rangle + g(x^k, y^{k,t}; \psi) - g(x^k, y^*(x^k); \psi)] + \\
&\quad 2\beta_{b,k}[g(x^k, y^*(x^k); \psi) - g(x^k, y^*_{x^k,\psi}; \psi)],
\end{aligned}
$$

where (a) is by Lemma 1, (b) is by choosing $p \geq \frac{1}{2}$, and, (c) is by individual convexity of $g(x, y; \psi)$ such that $-\langle y^{k,t} - y^*(x^k), h_g^{k,t}\rangle + g(x^k, y^{k,t}; \psi) - g(x^k, y^*(x^k); \psi) \leq 0$ and recalling that $C_k = \min\{\frac{1}{2pL_g}, \beta_{b,k}\} \leq \beta_{k,t}$ by Lemma 1. Take expectation conditioned on $\mathcal{F}'_{k,t}$ and note that $\mathbb{E}[h_g^{k,t}|\mathcal{F}'_{k,t}] = \nabla_y g(x^k, y^{k+1})$, $\mathbb{E}[g(x^k, y^{k,t}; \psi)|\mathcal{F}'_{k,t}] = g(x^k, y^{k,t})$, and $\mathbb{E}[g(x^k, y^*(x^k); \psi)] = g(x^k, y^*(x^k))$:

$$
\begin{aligned}
\mathbb{E}[\|y^{k,t+1} - y^*(x^k)\|^2|\mathcal{F}'_{k,t}] &\leq \|y^{k,t} - y^*(x^k)\|^2 + 2C_k[-\langle y^{k,t} - y^*(x^k), \nabla_y g(x^k, y^{k+1})\rangle + \\
&\quad g(x^k, y^{k,t}) - g(x^k, y^*(x^k))] + 2\beta_{b,k}\sigma_g^2. \quad (27)
\end{aligned}
$$

Now, based on the strong convexity of $g$ w.r.t. $y$,

$$
-\langle y^{k,t} - y^*(x^k), \nabla_y g(x^k, y^{k+1})\rangle + g(x^k, y^{k,t}) - g(x^k, y^*(x^k)) \leq \frac{\mu_g}{2}\|y^{k,t} - y^*(x^k)\|^2,
$$

we can further obtain (by taking total expectations of (27) and using strong-convexity):

$$
\mathbb{E}[\|y^{k,t+1} - y^*(x^k)\|^2] \leq (1 - \mu_g C_k)\mathbb{E}[\|y^{k,t} - y^*(x^k)\|^2] + 2\beta_{b,k}\sigma_g^2.
$$

Solve this recursively from $t = 0$ to $t = T - 1$ (recall $T$ is the total number of lower-level steps, $y^{k,0} = y^k$ and $y^{k+1} = y^{k,T}$):

$$
\begin{aligned}
\mathbb{E}[\|y^{k+1} - y^*(x^k)\|^2] &\leq (1 - \mu_g C_k)^T \mathbb{E}[\|y^k - y^*(x^k)\|^2] + 2\beta_{b,k}\sigma_g^2 \sum_{j=0}^{T-1}(1 - \mu_g C_k)^j \\
&\leq (1 - \mu_g C_k)^T \mathbb{E}[\|y^k - y^*(x^k)\|^2] + 2\beta_{b,k}T\sigma_g^2.
\end{aligned}
$$

$\square$

Lemma 6 is based on the standard bounded variance assumption $\mathbb{E}_\psi[\|\nabla_y g(x, y^*(x); \psi) - \nabla_y g(x, y^*(x))\|^2] \leq \sigma_g^2, \forall x, y$ in the bi-level optimization literature [19, 4]. Lemma 6 will be used in the proof of Theorem 4.

**Lemma 6.** *Suppose Assumptions 3, 5 and the bounded variance assumption* (16) *hold. Suppose that* $p \geq \max\{\frac{\mu_g}{\mu_g+L_g}, \frac{\mu_g+L_g}{4L_g}\}$, $\beta_{b,0} \leq \min\{\frac{2}{\mu_g+L_g}, \frac{\mu_g+L_g}{2\mu_g L_g}, \frac{1}{2pL_g - \frac{2\mu_g L_g}{\mu_g+L_g}}\}$. *Then step size of the form 11 achieves the following:*

$$
\mathbb{E}[\|y^{k+1} - y^*(x^k)\|^2] \leq (\frac{\beta_{b,k}}{C_k} - \frac{2\mu_g L_g}{\mu_g + L_g}\beta_{b,k})^T \mathbb{E}[\|y^k - y^*(x^k)\|^2] + T\beta_{b,k}^2\sigma_g^2,
$$

*where* $C_k = \min\{\frac{1}{2pL_g}, \beta_{b,k}\}$.

*Proof.* Similar to the proof of Lemma 5, we can start with

$$\|y^{k,t+1} - y^*(x^k)\|^2 = \|y^{k,t} - y^*(x^k)\|^2 - 2\beta_{k,t}\langle y^{k,t} - y^*(x^k), h_g^{k,t}\rangle + \beta_{k,t}^2\|h_g^{k,t}\|^2.$$

Divide both sides by $\beta_{k,t}$

$$\frac{\|y^{k,t+1} - y^*(x^k)\|^2}{\beta_{k,t}} = \frac{\|y^{k,t} - y^*(x^k)\|^2}{\beta_{k,t}} - 2\langle y^{k,t} - y^*(x^k), h_g^{k,t}\rangle + \beta_{k,t}\|h_g^{k,t}\|^2.$$

Then use the facts that $\beta_{k,t} \leq \beta_{b,k}$ and $\beta_{k,t} \geq C_k$,

$$\frac{\|y^{k,t+1} - y^*(x^k)\|^2}{\beta_{b,k}} \leq \frac{\|y^{k,t} - y^*(x^k)\|^2}{C_k} - 2\langle y^{k,t} - y^*(x^k), h_g^{k,t}\rangle + \beta_{b,k}\|h_g^{k,t}\|^2.$$

Next, take expectation conditioned on the Filtration $\mathcal{F}'_{k,t}$ up to and including $x^k$ and $y^{k,t}$

$$\frac{1}{\beta_{b,k}}\mathbb{E}[\|y^{k,t+1} - y^*(x^k)\|^2|\mathcal{F}'_{k,t}] \leq \frac{1}{C_k}\mathbb{E}[\|y^{k,t} - y^*(x^k)\|^2|\mathcal{F}'_{k,t}] - 2\langle y^{k,t} - y^*(x^k), \nabla g(x^k, y^{k,t})\rangle$$
$$+ \beta_{b,k}\mathbb{E}[\|h_g^{k,t}\|^2|\mathcal{F}'_{k,t}]$$
$$= \frac{1}{C_k}\mathbb{E}[\|y^{k,t} - y^*(x^k)\|^2|\mathcal{F}'_{k,t}] - 2\langle y^{k,t} - y^*(x^k), \nabla g(x^k, y^{k,t})\rangle$$
$$+ \beta_{b,k}\mathbb{E}[\|h_g^{k,t} - \nabla g(x^k, y^{k,t}) + \nabla g(x^k, y^{k,t})\|^2|\mathcal{F}'_{k,t}]$$
$$\overset{(a)}{\leq} \frac{1}{C_k}\mathbb{E}[\|y^{k,t} - y^*(x^k)\|^2|\mathcal{F}'_{k,t}] - 2\langle y^{k,t} - y^*(x^k), \nabla g(x^k, y^{k,t})\rangle$$
$$+ \beta_{b,k}\sigma_g^2 + \beta_{b,k}\|\nabla g(x^k, y^{k,t})\|^2,$$

where (a) is by the bounded variance assumption (16). Based on strong-convexity of $g$ [34, Theorem 2.1.11], we have

$$\frac{1}{\beta_{b,k}}\mathbb{E}[\|y^{k,t+1} - y^*(x^k)\|^2|\mathcal{F}'_{k,t}] \leq \frac{1}{C_k}\|y^{k,t} - y^*(x^k)\|^2 - \frac{2\mu_g L_g}{\mu_g + L_g}\|y^{k,t} - y^*(x^k)\|^2$$
$$- \frac{2}{\mu_g + L_g}\|\nabla g(x^k, y^{k,t})\|^2 + \beta_{b,k}\sigma_g^2 + \beta_{b,k}\|\nabla g(x^k, y^{k,t})\|^2$$

Multiply by $\beta_{b,k}$ in both sides, group terms, and take total expectations to reach

$$\mathbb{E}[\|y^{k,t+1} - y^*(x^k)\|^2] \leq (\frac{\beta_{b,k}}{C_k} - \frac{2\mu_g L_g}{\mu_g + L_g}\beta_{b,k})\mathbb{E}[\|y^{k,t} - y^*(x^k)\|^2] + \beta_{b,k}^2\sigma_g^2$$
$$+ \beta_{b,k}(\beta_{b,k} - \frac{2}{\mu_g + L_g})\|\nabla g(x^k, y^{k,t})\|^2$$
$$\overset{(b)}{\leq} (\frac{\beta_{b,k}}{C_k} - \frac{2\mu_g L_g}{\mu_g + L_g}\beta_{b,k})\mathbb{E}[\|y^{k,t} - y^*(x^k)\|^2] + \beta_{b,k}^2\sigma_g^2,$$

where in (b) we have chosen $\beta_{b,k} \leq \beta_{b,0} \leq \frac{2}{\mu_g+L_g}, \forall k$. Solving this recursion similar to Lemma 5, we obtain:

$$\mathbb{E}[\|y^{k+1} - y^*(x^k)\|^2] \leq (\frac{\beta_{b,k}}{C_k} - \frac{2\mu_g L_g}{\mu_g + L_g}\beta_{b,k})^T\mathbb{E}[\|y^k - y^*(x^k)\|^2] + T\beta_{b,k}^2\sigma_g^2. \qquad (28)$$

In (28), we require $\frac{\beta_{b,k}}{C_k} - \frac{2\mu_g L_g}{\mu_g+L_g}\beta_{b,k} \geq 0$ (recall $C_k = \min\{\frac{1}{2pL_g}, \beta_{b,k}\} \leq \beta_{k,t} \leq \beta_{b,k}$), this is equivalent to $C_k \leq \frac{\mu_g+L_g}{2\mu_g L_g}$. In case of $C_k = \frac{1}{2pL_g}$, we choose $p \geq \frac{\mu_g}{\mu_g+L_g}$ (to avoid contradictions, we also choose $p \geq \frac{\mu_g+L_g}{4L_g}$). In case of $C_k = \beta_{b,k}$, we choose $\beta_{b,k} \leq \beta_{b,0} \leq \frac{\mu_g+L_g}{2\mu_g L_g}$.

We also require $\frac{\beta_{b,k}}{C_k} - \frac{2\mu_g L_g}{\mu_g+L_g}\beta_{b,k} \leq 1$. In case of $C_k = \beta_{b,k}$, this is equivalent to $\frac{2\mu_g L_g}{\mu_g+L_g}\beta_{b,k} \geq 0$, which is satisfied by all $\beta_{b,k}$. In case of $C_k = \frac{1}{2pL_g}$, we choose $\beta_{b,k} \leq \beta_{b,0} \leq \frac{1}{2pL_g - \frac{2\mu_g L_g}{\mu_g+L_g}}$. Puting everything together, we have $p \geq \max\{\frac{\mu_g}{\mu_g+L_g}, \frac{\mu_g+L_g}{4L_g}\}$, and $\beta_{b,0} \leq \min\{\frac{2}{\mu_g+L_g}, \frac{\mu_g+L_g}{2\mu_g L_g}, \frac{1}{2pL_g - \frac{2\mu_g L_g}{\mu_g+L_g}}\}$. $\qquad \square$

## A.2 Single-level Convex Proofs

### A.2.1 Proof of Theorem 1

The proof of Theorem 1 is similar to the standard proof of decaying-step SGD (GD) that can be found in e.g. Beck [1]. Here, we give the proof for completeness.

$$\|x^{k+1} - x^*\|^2 = \|x^k - x^*\|^2 - 2\gamma_k\langle x^k - x^*, \nabla f_i(x^k)\rangle + \gamma_k^2\|\nabla f_i(x^k)\|^2$$

$$\overset{(a)}{\leq} \|x^k - x^*\|^2 - 2\gamma_k(f_i(x^k) - f_i(x^*)) + \gamma_k^2\|\nabla f_i(x^k)\|^2$$

$$\overset{(b)}{\leq} \|x^k - x^*\|^2 - 2\gamma_k(f_i(x^k) - f_i(x^*)) + \gamma_{b,k}^2 G^2,$$

where (a) is by individual convexity of $f_i$, and (b) is by Assumption 2. Take conditional expectation and assume that $\gamma_k$ is independent of sample $k$

$$\mathbb{E}[\|x^{k+1} - x^*\|^2|x^k] \overset{(c)}{\leq} \|x^k - x^*\|^2 - 2E[\gamma_k|x^k](f(x^k) - f(x^*)) + \gamma_{b,k}^2 G^2$$

$$\overset{(d)}{\leq} \|x^k - x^*\|^2 - 2\gamma_{l,k}(f(x^k) - f(x^*)) + \gamma_{b,k}^2 G^2,$$

where (c) is by independence of $\gamma_k$ and $x^k$, and (d) is because $\gamma_{l,k} \leq \gamma_k, \forall k$. Take total expectation and rearrange

$$2\gamma_{l,k}\mathbb{E}[f(x^k) - f(x^*)] \leq \mathbb{E}[\|x^k - x^*\|^2] - E[\|x^{k+1} - x^*\|^2] + \gamma_{b,k}^2 G^2$$

Using the fact that $\gamma_{l,K-1} \leq \gamma_{l,k} \quad \forall k \in [K-1]$ and set $\gamma_{b,k} = \frac{\gamma_0}{\sqrt{k+1}}$, we obtain

$$2\gamma_{l,K-1}\mathbb{E}[f(x^k) - f(x^*)] \leq \mathbb{E}[\|x^k - x^*\|^2] - E[\|x^{k+1} - x^*\|^2] + \frac{\gamma_0^2}{k+1}G^2$$

Summing over $k = 0$ to $k = K - 1$ we obtain

$$2\gamma_{l,K-1}\frac{1}{K}\sum_{k=0}^{K-1}\mathbb{E}[f(x^k) - f(x^*)] \leq \frac{\|x^0 - x^*\|^2 - \mathbb{E}[\|x^K - x^*\|^2]}{K} + \gamma_0^2 G^2 \frac{1}{K}\sum_{k=0}^{K-1}\frac{1}{k+1}$$

$$\leq \frac{\|x^0 - x^*\|^2}{K} + \frac{\gamma_0^2 G^2 \log(K)}{K}$$

Define $\bar{x} = \frac{1}{K}\sum_{k=0}^{K-1} x^k$, apply Jensen's inequality and rearrange

$$\mathbb{E}[f(\bar{x}^K) - f(x^*)] \leq \frac{\|x^0 - x^*\|^2}{2\gamma_{l,K-1}K} + \frac{\gamma_0^2 G^2 \log(K)}{2\gamma_{l,K-1}K}.$$

## A.3 Proof of Theorem 2

The approach of Theorem 2 is similar to [16, Theorem 3.2]. The crucial difference is that the step size in [16, Theorem 3.2] is constant ($\gamma_k = \frac{1}{L_{\max}}$) for $k \leq 4\lceil\kappa\rceil$, whereas for envelope-type step size it is of the form:

$$\gamma_k = \min\{\max\{\gamma_{l,k}, \tilde{\gamma}_k\}, \gamma_{b,k}\}, \quad \gamma_{l,k} = \min\{w, \gamma_{b,k}\},$$

where $\bar{\gamma}_k$ can be (e.g. in the case of SPSB ):

$$\bar{\gamma}_k = \min\{\frac{f_{i_k}(x^k) - l_{i_k}^*}{c_k\|\nabla f_{i_k}(x^k)\|^2}, \frac{\gamma_{b,0}}{k+1}\}, \quad w = \frac{1}{2c_k L_{\max}}.$$

The proof of Theorem 2 suggests that the step size can be either $\frac{f_{i_k}(x^k) - l_{i_k}^*}{c_k\|\nabla f_{i_k}(x^k)\|^2}$ or $\frac{\gamma_{b,0}}{k+1}$ depending on their magnitudes for $k \leq k_0 - 1$ ($k_0 = \max\{1, \lceil\gamma_0/w\rceil - 1\}$). After $k_0$ iterations, the step size is $\gamma_k = \frac{\gamma_{b,0}}{k+1}$. This finding is numerically confirmed by the experimental results in Section B.

To proceed with the proof, we have:

$$\|x^{k+1} - x^*\|^2 = \|x^k - \gamma_k \nabla f_i(x^k) - x^*\|^2$$

$$= \|x^k - x^*\|^2 - 2\gamma_k \langle \nabla f_i(x^k), x^k - x^* \rangle + \gamma_k^2 \|\nabla f_i(x^k)\|^2$$

$$\overset{(a)}{\leq} \|x^k - x^*\|^2 - 2\gamma_k \langle \nabla f_i(x^k), x^k - x^* \rangle + \gamma_{b,k}^2 \|\nabla f_i(x^k)\|^2,$$

where (a) is because $\gamma_k \leq \gamma_{b,k}, \forall k$. Take conditional expectations

$$\mathbb{E}[\|x^{k+1} - x^*\|^2 | x^k] \leq \|x^k - x^*\|^2 - 2\mathbb{E}[\gamma_k | x^k]\langle \nabla f(x^k), x^k - x^* \rangle + \gamma_{b,k}^2 \mathbb{E}[\|\nabla f_i(x^k)\|^2 | x^k]$$

$$\overset{(b)}{\leq} \|x^k - x^*\|^2 - \mu \mathbb{E}[\gamma_k | x^k]\|x^k - x^*\|^2 - 2\mathbb{E}[\gamma_k | x^k][f(x^k) - f(x^*)] + \gamma_{b,k}^2 G^2$$

$$\overset{(c)}{\leq} \|x^k - x^*\|^2 - \mu \gamma_{l,k}\|x^k - x^*\|^2 - 2\gamma_{l,k}[f(x^k) - f(x^*)] + \gamma_{b,k}^2 G^2$$

where (b) is by bounded gradients assumption and strong convexity of $f$. Take total expectation and rearrange

$$2\mathbb{E}[f(x^k) - f(x^*)] \leq \frac{(1 - \mu \gamma_{l,k})\mathbb{E}[\|x^k - x^*\|^2] - \mathbb{E}[\|x^{k+1} - x^*\|^2]}{\gamma_{l,k}} + \frac{\gamma_{b,k}^2 G^2}{\gamma_{l,k}}$$

Choose $k_0 = \max\{1, \lceil \gamma_0/\omega \rceil - 1\}$, then for $\forall k$ s.t. $k \geq k_0$, we have $\gamma_{l,k} = \min\{\omega, \gamma_{b,k}\} = \gamma_{b,k} = \frac{\gamma_0}{k+1}$, which means $\gamma_k = \frac{\gamma_0}{k+1}$ after $k_0$ steps. Within the first $k_0$ steps, the step size is $\gamma_k = \min\{\omega, \gamma_{b,k}\}$. Hence, for $k \geq k_0$ we have

$$2\mathbb{E}[f(x^k) - f(x^*)] \leq \frac{(1 - \mu \gamma_{l,k})\mathbb{E}[\|x^k - x^*\|^2] - \mathbb{E}[\|x^{k+1} - x^*\|^2]}{\gamma_{l,k}} + \frac{\gamma_0 G^2}{k+1}$$

Now, sum from $k = k_0$ to $K - 1$

$$2 \sum_{k=k_0}^{K-1} \mathbb{E}[f(x^k) - f(x^*)] \leq \sum_{k=k_0}^{K-1} \frac{(1 - \mu \gamma_{l,k})\mathbb{E}[\|x^k - x^*\|^2] - \mathbb{E}[\|x^{k+1} - x^*\|^2]}{\gamma_{l,k}} + \sum_{k=k_0}^{K-1} \frac{\gamma_0 G^2}{k+1} \tag{29}$$

For the first term in (29), call it $A$, we expand it as

$$A = \sum_{k=k_0+1}^{K-1} \mathbb{E}[\|x^k - x^*\|^2](\frac{1}{\gamma_{l,k}} - \frac{1}{\gamma_{l,k-1}} - \mu) + \mathbb{E}[\|x_{k_0} - x^*\|^2](\frac{1}{\gamma_{l,k_0}} - \mu) - \frac{\mathbb{E}[\|x_K - x^*\|^2]}{\gamma_{l,K-1}}$$

$$\leq \sum_{k=k_0+1}^{K-1} \mathbb{E}[\|x^k - x^*\|^2](\frac{k+1}{\gamma_0} - \frac{k}{\gamma_0} - \mu) + \mathbb{E}[\|x_{k_0} - x^*\|^2](\frac{k_0+1}{\gamma_0} - \mu)$$

$$\overset{(d)}{\leq} \mathbb{E}[\|x_{k_0} - x^*\|^2]\mu k_0.$$

where (d) is because $\gamma_0 \geq \frac{1}{\mu}$, we have $\frac{k+1}{\gamma_0} - \frac{k}{\gamma_0} - \mu \leq 0, \forall k$ and $\frac{k_0+1}{\gamma_0} - \mu \leq \mu k_0$. For the second term in (29), call it B, we have

$$B = \sum_{k=k_0}^{K-1} \frac{\gamma_0 G^2}{k+1} \leq \gamma_0 G^2 \int_{k_0}^{K-1} \frac{1}{x+1} dx = \gamma_0 G^2[\log(K) - \log(k_0 + 1)] \leq \gamma_0 G^2 \log K$$

Putting $A$ and $B$ together, we obtain

$$\frac{1}{K - k_0} \sum_{k=k_0}^{K-1} \mathbb{E}[f(x^k) - f(x^*)] \leq \frac{\mathbb{E}[\|x^{k_0} - x^*\|^2]\mu k_0}{2(K - k_0)} + \frac{\gamma_0 G^2 \log(K)}{2(K - k_0)} \tag{30}$$

Within the first $k_0 - 1$ iterations, similarly to the above, we have

$$\mathbb{E}[\|x^{k+1} - x^*\|] \leq (1 - \mu \gamma_{l,k})\mathbb{E}[\|x^k - x^*\|^2] - 2\gamma_{l,k}\mathbb{E}[f(x^k) - f(x^*)] + \gamma_{b,k}^2 G^2$$

$$\leq (1 - \mu \gamma_{l,k})\mathbb{E}[\|x^k - x^*\|^2] + \gamma_{b,k}^2 G^2.$$

For the first $k \leq k_0 - 1$ iterations, $\gamma_{l,k} = \omega$ where $\omega\mu < 1$; thus, we obtain the following

$$\mathbb{E}[\|x^{k+1} - x^*\|] \leq (1 - \mu\omega)E[\|x^k - x^*\|^2] + \gamma_{b,k}^2 G^2.$$

Solve this recursively,

$$
\begin{aligned}
\mathbb{E}[\|x^{k_0} - x^*\|^2] &\leq (1 - \mu\omega)^{k_0}\|x^0 - x^*\|^2 + \sum_{k=0}^{k_0-1}(1-\mu\omega)^{k_0-k-1}\frac{\gamma_0^2 G^2}{(k+1)^2} \\
&\leq (1-\mu\omega)^{k_0}\|x^0 - x^*\|^2 + \sum_{k=0}^{k_0-1}\frac{\gamma_0^2 G^2}{(k+1)^2} \\
&\leq (1-\mu\omega)^{k_0}\|x^0 - x^*\|^2 + \gamma_0^2 G^2 \int_0^{k_0-1}\frac{1}{(1+x)^2}dx \\
&= (1-\mu\omega)^{k_0}\|x^0 - x^*\|^2 + \gamma_0^2 G^2(1 - \frac{1}{k_0}) \\
&\leq (1-\mu\omega)^{k_0}\|x^0 - x^*\|^2 + \gamma_0^2 G^2. \quad (31)
\end{aligned}
$$

Putting this into (30), we obtain

$$\frac{1}{K-k_0}\sum_{k=k_0}^{K-1}\mathbb{E}[f(x^k) - f(x^*)] \leq \frac{\mu k_0}{2(K-k_0)}\{\exp(-k_0\mu\omega)\|x_0 - x^*\|^2 + \gamma_0^2 G^2\} + \frac{\gamma_0 G^2 \log K}{2(K-k_0)}.$$

Define $\bar{x}_K = \frac{1}{K-k_0}\sum_{k=k_0}^{K-1}x^k$, then by Jensen's inequality we have

$$\mathbb{E}[f(\bar{x}_K) - f(x^*)] \leq \frac{\mu k_0}{2(K-k_0)}\{\exp(-k_0\mu\omega)\|x_0 - x^*\|^2 + \gamma_0^2 G^2\} + \frac{\gamma_0 G^2 \log K}{2(K-k_0)},$$

where $k_0 = \max\{1, \lceil\gamma_0/\omega\rceil - 1\}$ and $\gamma_0 \geq \frac{1}{\mu}$.

## A.4 Bi-level Proofs

### A.4.1 Proof of Theorem 3

Start with Lemma 4:

$$E[V^{k+1}] \leq \mathbb{E}[V^k] - \frac{\alpha_{l,k}}{2}\mathbb{E}[\|\nabla F(x^k)\|^2] + (\alpha_{b,k}L_f^2 + 2)\mathbb{E}[\|y^{k+1} - y^*(x^k)\|^2]+$$

$$\alpha_{b,k}B^2 + (2L_y^2\alpha_{b,k}^2 + \frac{L_F\alpha_{b,k}^2}{2})\tilde{\sigma}_f^2 - \mathbb{E}[\|y^k - y^*(x^k)\|^2].$$

We substitute the result of Lemma 5 for the expression $\mathbb{E}[\|y^{k+1} - y^*(x^k)\|^2]$,

$$E[V^{k+1}] \leq \mathbb{E}[V^k] - \frac{\alpha_{l,k}}{2}\mathbb{E}[\|\nabla F(x^k)\|^2] + [(\alpha_{b,k}L_f^2 + 2)(1 - \mu_g C_k)^T - 1]\mathbb{E}[\|y^k - y^*(x^k)\|^2]+$$

$$2\alpha_{b,k}\beta_{b,k}TL_f^2\sigma_g^2 + 4\beta_{b,k}T\sigma_g^2 + \alpha_{b,k}B^2 + (2L_y^2 + \frac{L_F}{2})\alpha_{b,k}^2\tilde{\sigma}_f^2$$

$$\overset{(a)}{\leq} \mathbb{E}[V^k] - \frac{\alpha_{l,k}}{2}\mathbb{E}[\|\nabla F(x^k)\|^2] + [(\alpha_{b,0}L_f^2 + 2)(1 - \mu_g C_{K-1})^T - 1]\mathbb{E}[\|y^k - y^*(x^k)\|^2]+$$

$$2\alpha_{b,k}\beta_{b,k}TL_f^2\sigma_g^2 + 4\beta_{b,k}T\sigma_g^2 + \alpha_{b,k}B^2 + (2L_y^2 + \frac{L_F}{2})\alpha_{b,k}^2\tilde{\sigma}_f^2$$

$$\overset{(b)}{\leq} E[V^k] - \frac{\alpha_{l,k}}{2}E[\|\nabla F(x^k)\|^2] + 2\alpha_{b,k}\beta_{b,k}TL_f^2\sigma_g^2 + 4\beta_{b,k}T\sigma_g^2 + \alpha_{b,k}B^2 + (2L_y^2 + \frac{L_F}{2})\alpha_{b,k}^2\tilde{\sigma}_f^2,$$

where (a) is by $\alpha_{b,0} \geq \alpha_{b,k}, C_{K-1} \leq C_k, \forall k$, hence $(\alpha_{b,0}L_f^2)(1 - \mu_g C_{K-1})^T \geq (\alpha_{b,k}L_f^2)(1 - \mu_g C_k)^T$ (recall that $C_k = \min\{\frac{1}{2pL_g}, \beta_{b,k}\}$ in Lemma 1); (b) is by $T \geq \frac{\log[\alpha_{b,0}L_f^2 + 2]}{-\log(1-\mu C_{K-1})}$, which implies that $(\alpha_{b,0}L_f^2 + 2)(1 - \mu_g C_{K-1})^T \leq 1$. Now, rearrange and use the fact that $\alpha_{l,K-1} \leq \alpha_{l,k}$,

$$\alpha_{l,K-1}\mathbb{E}[\|\nabla F(x^k)\|^2] \leq 2\mathbb{E}[V^k] - 2\mathbb{E}[V^{k+1}] + 4\alpha_{b,k}\beta_{b,k}TL_f^2\sigma_g^2 + 8\beta_{b,k}T\sigma_g^2 + 2\alpha_{b,k}B^2 + (4L_y^2 + L_F)\alpha_{b,k}^2\tilde{\sigma}_f^2.$$

Then sum over $k = 0$ to $K - 1$:

$$\frac{1}{K}\sum_{k=0}^{K-1}\mathbb{E}[\|\nabla F(x^k)\|^2] \leq \frac{2V^0}{\alpha_{l,K-1}K} + \frac{4TL_f^2\sigma_g^2}{\alpha_{l,K-1}K}\sum_{k=0}^{K-1}\alpha_{b,k}\beta_{b,k} + \frac{8T\sigma_g^2}{\alpha_{l,K-1}K}\sum_{k=0}^{K-1}\beta_{b,k} + \frac{2B^2}{\alpha_{l,K-1}K}\sum_{k=0}^{K-1}\alpha_{b,k} +$$

$$\frac{(4L_y^2 + L_F)\tilde{\sigma}_f^2}{\alpha_{l,K-1}K}\sum_{k=0}^{K-1}\alpha_{b,k}^2$$

$$\stackrel{(c)}{=} \frac{2V^0}{\alpha_{l,K-1}K} + \frac{4TL_f^2\sigma_g^2\alpha_{b,0}\beta_{b,0}}{\alpha_{l,K-1}K}\sum_{k=0}^{K-1}\frac{1}{(k+1)^{1.5}} + \frac{8T\sigma_g^2\beta_{b,0}}{\alpha_{l,K-1}K}\sum_{k=0}^{K-1}\frac{1}{k+1} +$$

$$+ \frac{2B^2\alpha_{b,0}}{\alpha_{l,K-1}K}\sum_{k=0}^{K-1}\frac{1}{\sqrt{k+1}} + \frac{(4L_y^2 + L_F)\tilde{\sigma}_f^2\alpha_{b,0}^2}{\alpha_{l,K-1}K}\sum_{k=0}^{K-1}\frac{1}{k+1}$$

$$\stackrel{(d)}{\leq} \frac{2V^0}{\alpha_{l,0}\sqrt{K}} + \frac{8TL_f^2\sigma_g^2\alpha_{b,0}\beta_{b,0}}{\alpha_{l,0}\sqrt{K}} + \frac{8T\sigma_g^2\beta_{b,0}\log(K)}{\alpha_{l,0}\sqrt{K}} +$$

$$\frac{2B^2\alpha_{b,0}}{\alpha_{l,0}} + \frac{(4L_y^2 + L_F)\tilde{\sigma}_f^2\alpha_{b,0}^2\log(K)}{\alpha_{l,0}\sqrt{K}},$$

where we substituted $\alpha_{b,k} = \frac{\alpha_{b,0}}{\sqrt{k+1}}$ and $\beta_{b,k} = \frac{\beta_{b,0}}{k+1}$ in (c); (d) is based on $\sum_{k=0}^{K-1}\frac{1}{(k+1)^{1.5}} \leq 2$, $\sum_{k=0}^{K-1}\frac{1}{k+1} \leq \log(K)$, and $\sum_{k=0}^{K-1}\frac{1}{\sqrt{k+1}} \leq \sqrt{K}$. Recall that in Lemma 2, we have $L_f \sim \mathcal{O}(\kappa^2)$, $L_y \sim \mathcal{O}(\kappa)$, and $L_F \sim \mathcal{O}(\kappa^3)$. Also recall that $\alpha_{b,k} = \frac{\alpha_{b,0}}{\sqrt{k+1}}$ and $\alpha_{l,k} = \frac{\alpha_{l,0}}{\sqrt{k+1}}$, hence we choose $\alpha_{l,0} \sim \alpha_{b,0} \sim \mathcal{O}(\kappa^{-3})$, $T \sim \mathcal{O}(\kappa)$, and $\beta_{b,0} \sim \mathcal{O}(\kappa^{-2})$. Then we can obtain $\frac{1}{K}\sum_{k=0}^{K-1}\mathbb{E}[\|\nabla F(x^k)\|^2] \leq \mathcal{O}(\frac{\kappa^3}{\sqrt{K}} + \frac{\kappa^2\log(K)}{\sqrt{K}})$.

### A.4.2 Theorem 4 and its proof

**Theorem 4.** *Suppose $f$ and $g$ satisfy Assumptions 3, 4, and 5, and, learning-rate upper bounds $\alpha_{b,k} = \frac{\alpha_{b,0}}{\sqrt{k+1}}$ and $\alpha_{l,k} = \frac{\alpha_{l,0}}{\sqrt{k+1}}$ with $\alpha_{b,0}$ and $\alpha_{l,0}$ satisfying $\frac{1}{L_F+4L_y^2} \geq \frac{\alpha_{b,0}^2}{\alpha_{l,0}}$ and $\alpha_{l,0} \leq \alpha_{b,0}$. Further assume that $\alpha_k$ is independent of the stochastic hypergradient $h_f^k$. Then, under the Bounded-variance assumption in (16) with $p \geq \max\{\frac{\mu_g}{\mu_g+L_g}, \frac{\mu_g+L_g}{4L_g}\}$, $\beta_{b,0} \leq \min\{\frac{2}{\mu_g+L_g}, \frac{\mu_g+L_g}{2\mu_gL_g}, \frac{1}{2pL_g-\frac{2\mu_gL_g}{\mu_g+L_g}}\}$, and*

$$T \geq \frac{\log(\frac{3}{2}\alpha_{b,0}L_f^2+2)}{\min\{-\log(1-\frac{2\mu_gL_g}{\mu_g+L_g}\beta_{b,K-1}), -\log((2pL_g-\frac{2\mu_gL_g}{\mu_g+L_g})\beta_{b,0})\}}, \text{ BiSPS achieves the following rate:}$$

$$\frac{1}{K}\sum_{k=0}^{K-1}\mathbb{E}[\|\nabla F(x^k)\|^2] \leq \tilde{\mathcal{O}}(\frac{\kappa^3}{\sqrt{K}} + \frac{\kappa^2\log K}{\sqrt{K}}).$$

*Proof.* Start with Lemma 4:

$$E[V^{k+1}] \leq \mathbb{E}[V^k] - \frac{\alpha_{l,k}}{2}\mathbb{E}[\|\nabla F(x^k)\|^2] + (\alpha_{b,k}L_f^2 + 2)\mathbb{E}[\|y^{k+1} - y^*(x^k)\|^2] +$$

$$\alpha_{b,k}B^2 + (2L_y^2\alpha_{b,k}^2 + \frac{L_F\alpha_{b,k}^2}{2})\tilde{\sigma}_f^2 - \mathbb{E}[\|y^k - y^*(x^k)\|^2].$$

We substitute the result of Lemma 6 for the expression $\mathbb{E}[\|y^{k+1} - y^*(x^k)\|^2]$,

$$E[V^{k+1}] \leq \mathbb{E}[V^k] - \frac{\alpha_{l,k}}{2}\mathbb{E}[\|\nabla F(x^k)\|^2] + [(\alpha_{b,k}L_f^2 + 2)(\frac{\gamma_{b,k}}{C_k} - \frac{2\mu_gL_g}{\mu_g+L_g}\beta_{b,k})^T - 1]\mathbb{E}[\|y^k - y^*(x^k)\|^2] +$$

$$+ L_f^2T\sigma_g^2\alpha_{b,k}\beta_{b,k}^2 + 2T\sigma_g^2\beta_{b,k}^2 + \alpha_{b,k}B^2 + [2L_y^2 + \frac{L_F}{2}]\alpha_{b,k}^2\tilde{\sigma}_f^2$$

$$\stackrel{(a)}{\leq} E[V^k] - \frac{\alpha_{l,k}}{2}\mathbb{E}[\|\nabla F(x^k)\|^2] + L_f^2T\sigma_g^2\alpha_{b,k}\beta_{b,k}^2 + 2T\sigma_g^2\beta_{b,k}^2 + \alpha_{b,k}B^2 + [2L_y^2 + \frac{L_F}{2}]\alpha_{b,k}^2\tilde{\sigma}_f^2,$$

where in (a) we have chosen $T \geq \max\{\frac{\log(\alpha_{b,0}L_f^2+2)}{-\log(1-\frac{2\mu_g L_g}{\mu_g+L_g}\beta_{b,K-1})}, \frac{\log(\alpha_{b,0}^2 L_f^2+2)}{-\log(2pL_g-\frac{2\mu_g L_g}{\mu_g+L_g})\beta_{b,0}}\}$. This ensures

that $T \geq \frac{\log(\alpha_{b,0}L_f^2+2)}{-\log(\frac{\beta_{b,k}}{C_k}-\frac{2\mu_g L_g}{\mu_g+L_g}\beta_{b,k})}, \forall k$, which guarantees that $(\alpha_{b,k}L_f^2+2)(\frac{\beta_{b,k}}{C_k}-\frac{2\mu_g L_g}{\mu_g+L_g}\beta_{b,k})^T - 1 \leq 0$.
Now, rearrange and use the fact that $\alpha_{l,K-1} \leq \alpha_{l,k}$,

$$\alpha_{l,K-1}\mathbb{E}[\|\nabla F(x^k)\|^2] \leq 2\mathbb{E}[V^k] - 2\mathbb{E}[V^{k+1}] + 2\alpha_{b,k}\beta_{b,k}^2 T L_f^2 \sigma_g^2 + 4\beta_{b,k}^2 T\sigma_g^2 + 2\alpha_{b,k}B^2 + (4L_y^2+L_F)\alpha_{b,k}^2\tilde{\sigma}_f^2.$$

Sum over $k = 0$ to $k = K-1$:

$$\frac{1}{K}\sum_{k=0}^{K-1}\mathbb{E}[\|\nabla F(x^k)\|^2] \leq \frac{2V^0}{\alpha_{l,K-1}K} + \frac{2TL_f^2\sigma_g^2}{\alpha_{l,K-1}K}\sum_{k=0}^{K-1}\alpha_{b,k}\beta_{b,k}^2 + \frac{4T\sigma_g^2}{\alpha_{l,K-1}K}\sum_{k=0}^{K-1}\beta_{b,k}^2 +$$

$$\frac{2B^2}{\alpha_{l,K-1}K}\sum_{k=0}^{K-1}\alpha_{b,k} + \frac{(4L_y^2+L_F)\tilde{\sigma}_f^2}{\alpha_{l,K-1}K}\sum_{k=0}^{K-1}\alpha_{b,k}^2$$

$$\overset{(b)}{=} \frac{2V^0}{\alpha_{l,K-1}K} + \frac{2TL_f^2\sigma_g^2\alpha_{b,0}\beta_{b,0}^2}{\alpha_{l,K-1}K}\sum_{k=0}^{K-1}\frac{1}{(k+1)^{3/2}} + \frac{4T\sigma_g^2\beta_{b,0}^2}{\alpha_{l,K-1}K}\sum_{k=0}^{K-1}\frac{1}{k+1} +$$

$$\frac{2B^2\alpha_{b,0}}{\alpha_{l,K-1}K}\sum_{k=0}^{K-1}\frac{1}{(k+1)^{0.5}} + \frac{(4L_y^2+L_F)\tilde{\sigma}_f^2\alpha_{b,0}^2}{\alpha_{l,K-1}K}\sum_{k=0}^{K-1}\frac{1}{k+1}$$

$$\overset{(c)}{\leq} \frac{2V^0}{\alpha_{l,K-1}K} + \frac{2TL_f^2\sigma_g^2\alpha_{b,0}\beta_{b,0}^2}{\alpha_{b,0}\sqrt{K}} + \frac{4T\sigma_g^2\beta_{b,0}^2\log(K)}{\alpha_{b,0}\sqrt{K}} +$$

$$+ \frac{2B^2\alpha_{b,0}}{\alpha_{l,0}} + \frac{(4L_y^2+L_F)\tilde{\sigma}_f^2\alpha_{b,0}^2\log(K)}{\alpha_{l,0}\sqrt{K}},$$

where in (b) we have substituted $\alpha_{b,k} = \frac{\alpha_{b,0}}{\sqrt{k+1}}$ and $\beta_{b,k} = \frac{\beta_{b,0}}{\sqrt{k+1}}$, and (c) is by $\sum_{k=0}^{K-1}\frac{1}{(k+1)^{3/2}} \leq 2$, $\sum_{k=0}^{K-1}\frac{1}{k+1} \leq \log(K)$, and $\sum_{k=0}^{K-1}\frac{1}{(k+1)^{1/2}} \leq \sqrt{K}$. Similar to the proof of Theorem 3, we choose $\alpha_{l,0} \sim \alpha_{b,0} \sim \mathcal{O}(\kappa^{-3})$, $T \sim \mathcal{O}(\kappa)$, and $\beta_{b,0} \sim \mathcal{O}(\kappa^{-1})$ to obtain $\frac{1}{K}\sum_{k=0}^{K-1}\mathbb{E}[\|\nabla F(x^k)\|^2] \leq \mathcal{O}(\frac{\kappa^3}{\sqrt{K}} + \frac{\kappa^2\log(K)}{\sqrt{K}})$. $\qquad\square$

## B    Additional Experiment Results

This section is organized as follows. First, we discuss synthetic quadratics experiments. Second, we provide more details on the sensitivity of the algorithms to the choices of $\delta$ in (14), on the reset procedure, and on the search cost of BiSLS. Third, we compare the empirical performance of 1-sample vs 2-samples implementations of our algorithms for single-level convex and bi-level optimization. Some additional results for hyper-representation learning and data distillation experiments are also presented. We run 5 independent runs for all our experiments.

### B.1    Synthetic Quadratics

The experiments on quadratic functions are adapted from Loizou et al. [29]. The training objective is as follows:

$$f(x) = \frac{1}{2}(x-x_1^*)^T H_1(x-x_1^*) + \frac{1}{2}(x-x_2^*)^T H_2(x-x_2^*),$$

where $H_i$ ($i = 1, 2$) are positive definite. The optimal solutions $x_i^*$ ($i = 1, 2$) are generated randomly from a standard normal distribution. Specifically, $H_i$ is defined as follows:

$$H_i = O^T \cdot \text{Diag}(\log(1+\lambda_i)) \cdot O, \quad i = 1, 2,$$

where $O$ and $\lambda_i$ are taken from the spectral decomposition of $P^T P$, and $P$ is generated from the standard normal distribution. Figure 9 shows the convergence of various algorithms with different starting points. Interestingly, both SPSB with either 1 sample or 2 samples (1 sample for computing the gradient and the other for computing the step size) converge to the optimal solution (labelled with a star).

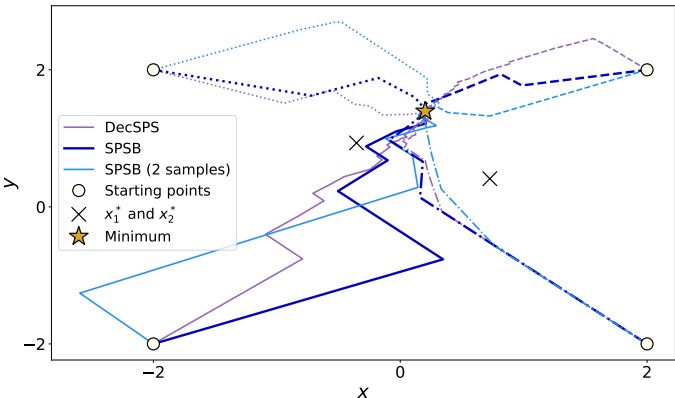

Figure 9: Iterate trajectories of different starting points for the synthetic quadratic experiments.

## B.2 Sensitivity of $\delta$, reset, and search cost

In this section, we discuss the effects of $\delta$ in (14), $\eta$ in Algorithm 2, and comment on the search cost of the options in the Reset Algorithm 2. Recall that the line-search condition (8) assumes that we can find a largest $\gamma_k \leq \gamma_{b,0}$ to satisfy it. However, in practice, we apply a backtracking procedure, i.e. $\gamma_k = \gamma_k * w, 0 < w < 1$, until $\gamma_k$ satisfies (8). Therefore, the found learning rate is not guaranteed to be the largest. Nonetheless, we assume that $\gamma_k$ is the largest to simplify our analysis given above (similar arguments apply to line-search at both upper and lower-level in the bi-level optimization). The experiments in this section are based on hyper-representation learning [42]. In this case, the objective of the induced bilevel-optimization problem can be written as:

$$\min_w F(w) = \frac{1}{2D_{X_1}}\|\tilde{f}(X_1; w)c^*(w) - Y_1\|^2$$

$$s.t. \quad c^*(w) = \underset{c}{\mathrm{argmin}} \frac{1}{2D_{X_2}}\|\tilde{f}(X_2; w)c - Y_2\|^2 + \frac{\lambda}{2}\|c\|^2,$$

where $(X_1, Y_1)$ and $(X_2, Y_2)$ are validation and training data sets with sizes $D_{X_1}$ and $D_{X_2}$, respectively; $\tilde{f}(\cdot; w)$ are the embedding layers of the model parameterized by $w$; and, $c$ is the classification layer. Moreover, we use conjugate gradient methods (CG) [17, 18] to solve the linear system when computing the hypergradient for hyper-representation learning experiments.

**Reset** While Algorithm 2 (reset) can be applied to both upper and lower-level problems, we focus our discussions here on the upper-level learning rate $(\alpha_k)$. This is because we empirically find it to be more critical for the convergence performance (see Figure 6a). As shown in Algorithm 2, reset has 3 options. Options 1, 2, and 3 search starting from $\alpha_{b,0}$, $\alpha_{k-1}$, and $\eta\alpha_{k-1}$, at iteration $k$ respectively. Option 1 has the highest search cost as it always starts from the same initial upper bound $(\alpha_{b,0})$. Option 2 ensures the monotonicity of the learning rate due to $\alpha_k \leq \alpha_{b,k} = \alpha_{k-1}$. Option 3 chooses the search starting point at iteration $k$ $(\alpha_{b,k})$ by multiplying the previous learning rate $(\alpha_{k-1})$ by a factor $\eta \geq 1$. As in the single-level convex case where monotonicity in the step size can potentially lead to slow convergence (see Figure 3), we again observe that monotonicity in the upper learning rate (i.e. option 2) leads to poorer performance when compared against options 1 or 3 as shown in Figure 10. Finally, we compare the performance of different $\eta$s in option 3 (note that $\eta = 1$ in option 3 is equivalent to option 2). We observe in Figure 11 that different $\eta$s perform equally well. This shows the robustness of our algorithm to the choice of $\eta$. As mentioned previously, the choice of option 3 over option 1 are due to 2 reasons: (a) reduced search cost; (b) provides an overall non-increasing and non-monotonic trend of upper bound $\alpha_{b,k}$. We discuss search cost of different $\eta$s in option 3 below.

**Search Cost** Based on the results in Figure 7b, we observe the use of option 3 in reset can significantly reduce the search cost for both BiSLS-Adam and BiSLS-SGD. Figure 10 further suggests that options 1 and 3 have nearly the same performance. Therefore, option 3 is an efficient algorithm that maintains good performance while reducing computation cost. Option 2 has the lowest search cost ($\sim 4$ rounds per iteration). However, its performance is not as good as option 1 or 3 as

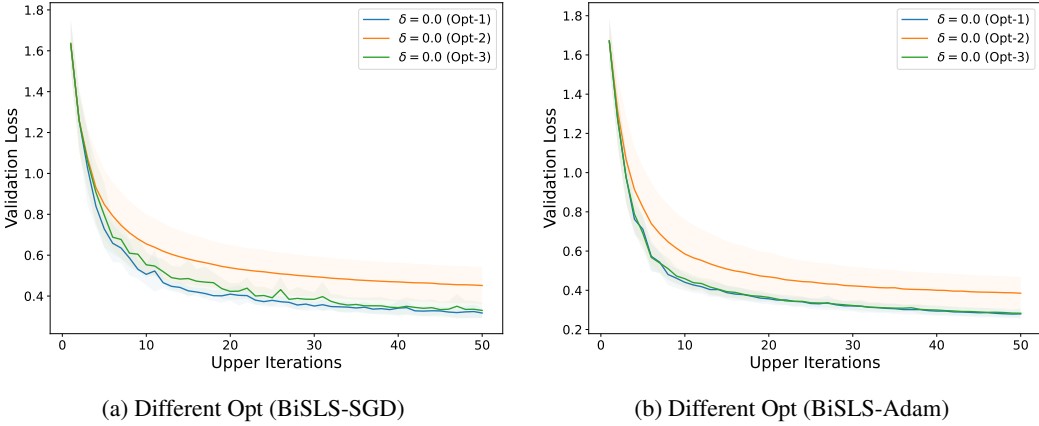

(a) Different Opt (BiSLS-SGD)    (b) Different Opt (BiSLS-Adam)

Figure 10: Validation loss against iterations with search options 1, 2, and 3 for the upper-level learning rate. The results for BiSLS-SGD and BiSLS-Adam are in (a) and (b), respectively. For the lower-level search, we fix it option 1 with $\beta_{b,0} = 100$. Results are based on hyper-representation learning.

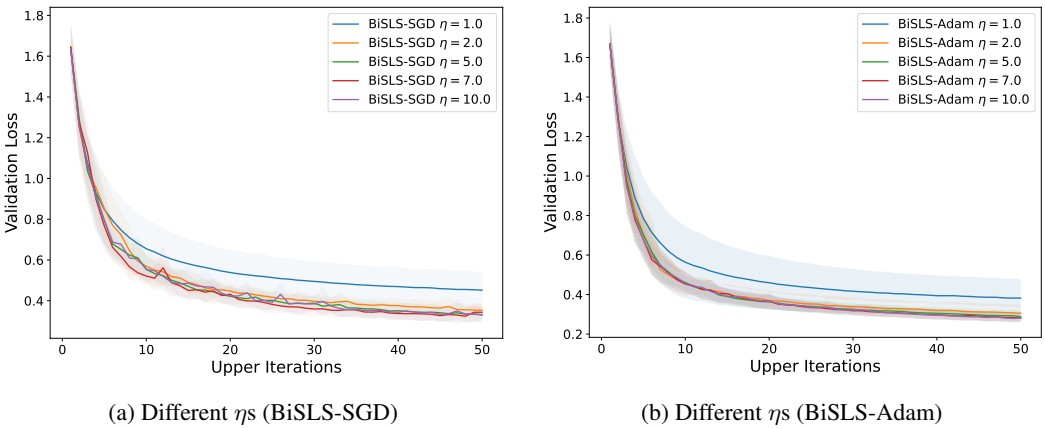

(a) Different $\eta$s (BiSLS-SGD)    (b) Different $\eta$s (BiSLS-Adam)

Figure 11: Validation loss against iterations for different $\eta$s based on reset option 3. Results for BiSLS-SGD are given in (a) and for BiSLS-Adam are given in (b). Note that $\eta = 1$ in reset option 3 is equivalent to reset option 2. For the lower-level search, we fix it option 1 with $\beta_{b,0} = 100$. Results are based on hyper-representation learning.

observed in Figure 10. Moreover, the average lower-level search cost is only 1 round per iteration when option 1 is used (see Figure 12).

**Sensitivity on $\delta$** As mentioned in Sec 2, due to the stochastic error in hypergradient computation, further complicated by the approximation error of $y^*(x)$ (see (14)), a learning rate is not guaranteed to be found in the bi-level case. Specifically, this is in contrast to the single-level convex problems. To avoid this, we introduce in (14) a $\delta$ slack to give some tolerance to such errors. Here, we give a thorough investigation of the effects of $\delta$ on performance. We vary its magnitude across 6 orders for both reset options 1 and 3 (see Algorithm 2 and discussions on reset above). We observe that despite a large difference on the magnitudes of $\delta$, they all share very similar performance for both BiSLS-SGD and BiSLS-Adam: see Figures 13 and 14. We summarize the key fins in this section as follows: ① **The option 3 in reset has good empirical performance (outperforms option 2) and is an effective way to reduce search cost (Figure 10, ??); ② BiSLS is highly robust to different choices of $\eta$ in option 3 and $\delta$ in (14) (Figure 11, 13, 14).**

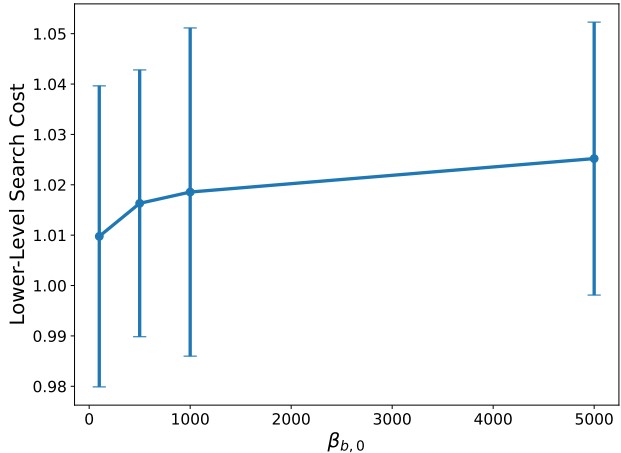

Figure 12: Lower-level search cost measured in the same way as upper-level against different lower-level search starting points ($\beta_{b,0}$). The lower-level search is done with option 1 (see above for discussions about these options).

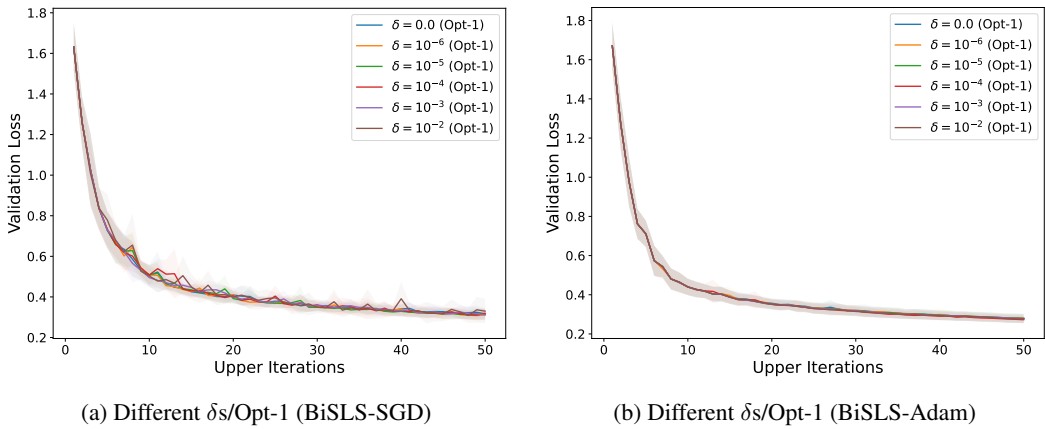

(a) Different $\delta$s/Opt-1 (BiSLS-SGD)  (b) Different $\delta$s/Opt-1 (BiSLS-Adam)

Figure 13: Validation loss against iterations for different $\delta$s based on reset option 1. Results for BiSLS-SGD (a) and for BiSLS-Adam (b). For the lower-level search, we fix it option 1 with $\beta_{b,0} = 100$. Results are based on hyper-representation learning.

### B.3 Data distillation objective and additional results

We let $\mathcal{L}_S(w)$ denote the loss evaluated on dataset $S$ with model weights $w$. The objective of *data distillation* can be expressed as a BO problem as follows:

$$D^* = \underset{D}{\operatorname{argmin}} \, \mathcal{L}_{\tilde{V}}(w^*(D)) \quad \text{s.t.} \quad w^*(D) = \underset{w}{\operatorname{argmin}} \, \mathcal{L}_D(w),$$

where $\tilde{V}$ is of the same size as $D$ and subsampled from the entire (original) dataset $V$. The solution $D^*$ is the distilled data, e.g. 9 MNIST digits each corresponding to a different label. In figure 15a, we show the performance of BiSPS for different values of $\alpha_{b,0}$ in comparison with BiSLS-SGD, and observe that BiSLS-SGD has better performance. In 15b, we show the results when we increase the number of lower-level iterations (T) from 20 to 50. As observed for $T = 20$ (in Figure 8), BiSLS-SGD here also outperforms a fine-tuned Adam or SGD.

### B.4 1-sample or 2-samples versions of algorithms for convex and bi-level optimization

We provide additional results to compare the performance of 1-sample and 2-samples (one for computing the gradient and the other for computing the step size) versions of our algorithms for

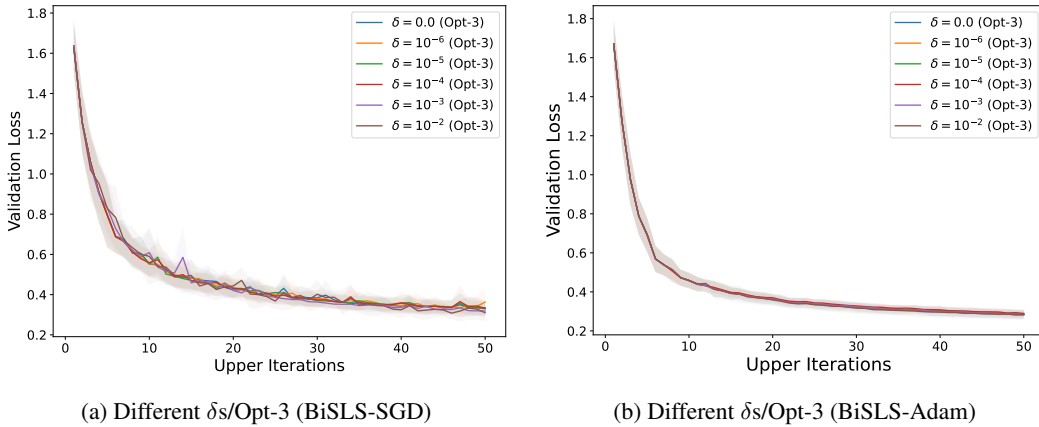

(a) Different $\delta$s/Opt-3 (BiSLS-SGD)      (b) Different $\delta$s/Opt-3 (BiSLS-Adam)

Figure 14: Validation loss against iterations for different $\delta$s based on reset option 3 ($\eta = 10$). Results for BiSLS-SGD (a) and for BiSLS-Adam (b). For the lower-level search, we fix it option 1 with $\beta_{b,0} = 100$. Results are based on hyper-representation learning.

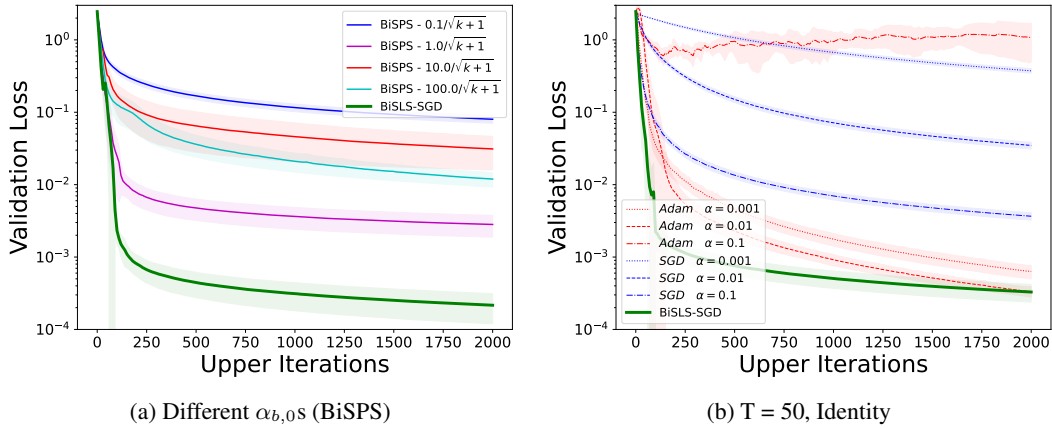

(a) Different $\alpha_{b,0}$s (BiSPS)      (b) T = 50, Identity

Figure 15: Validation loss against iterations. (a) Comparison between BiSPS with different $\alpha_{b,0}$s and BiSLS-SGD. (b) Comparison between BiSLS-SGD to fine-tuned Adam/SGD ($\beta_k$ fixed at $10^{-4}$). Inverse Hessian in (2) treated as the Identity [30] when computing the hypergradient. Recall that T is the total number of lower-level iterations and we have shown the results for $T = 20$ in Figure 8b.

SPSB and BiSPS used for single-level and bi-level optimization, respectively. In the single-level case (Figure 16), we observe that 2-samples SPSB performs just as well as 1-sample SPSB . Interestingly, we observe that their step sizes also follow a similar pattern. That is: an initial increase followed by a regime where $\gamma_k = \frac{f_{i_k}(x^k) - l^*_{i_k}}{c\|\nabla f_{i_k}(x^k)\|^2}$ is frequently used, and eventually changes to decaying-step SGD. This seems to also match with Theorem 2 where a transition point for SPSB ($k_0 = \max\{1, \lceil \gamma_0/\omega \rceil - 1\}, w = \frac{1}{2cL_{\max}}, \gamma_0 \geq \frac{1}{\mu}$) is predicted. At the same time, we also note that (perhaps, unsurprisingly) the 2-samples version seems to have a slightly more oscillatory behavior than the 1-sample version as shown in Figure 16. SLSB with either 1-sample or 2-samples also result in a similar performance and step size. Overall, despite the requirements of Theorems 1 and 2 for a 2-samples assumption, the empirical performance of 1-sample and 2-samples for either SPSB or SLSB appears to be very similar. Moving on to the bi-level case, recall that Theorems 3 and 4 require the 2-samples assumption (i.e., $\alpha_k$ independent of $h^k_f$) for the upper-level learning rate. We empirically verify this assumption with both hyper-representation learning and data distillation experiments. For hyper-representation learning experiments in Figure 17, BiSPS with either 1-sample or 2-samples for different values of $\alpha_{b,0}$ show similar performance. In fact, for $\alpha_{b,0} = 0.1$ we even observe that the 2-samples variant outperforms the 1-sample BiSPS. For data distillation experiments in Figure 18, the performances of 1-sample and 2-samples BiSPS are similar to each other when

$\alpha_{b,0} = 10.0$ or $\alpha_{b,0} = 50.0$. In general, the performance difference between 1-sample and 2-samples in the single-level or bi-level settings is small.

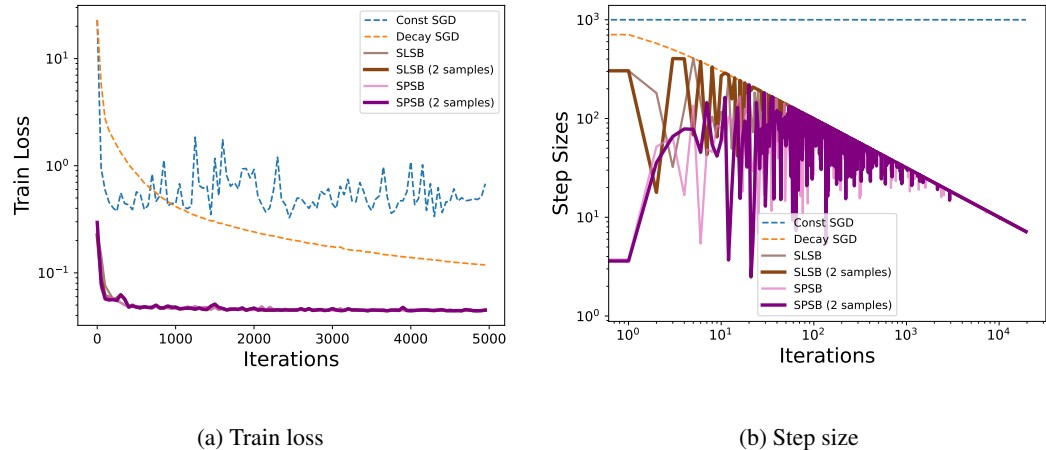

(a) Train loss

(b) Step size

Figure 16: Binary linear classification on w8a dataset using logistic loss [3]. Train loss (left) and step size (right) against iterations. We choose $\gamma_{b,0} = 1000$ for all algorithms. The upper bound for either SPSB or SLSB decays as $\gamma_{b,k} = \frac{\gamma_{b,0}}{\sqrt{k+1}}$. For decaying-step SGD, the learning rate schedule is $\frac{\gamma_{b,0}}{\sqrt{k+1}}$.

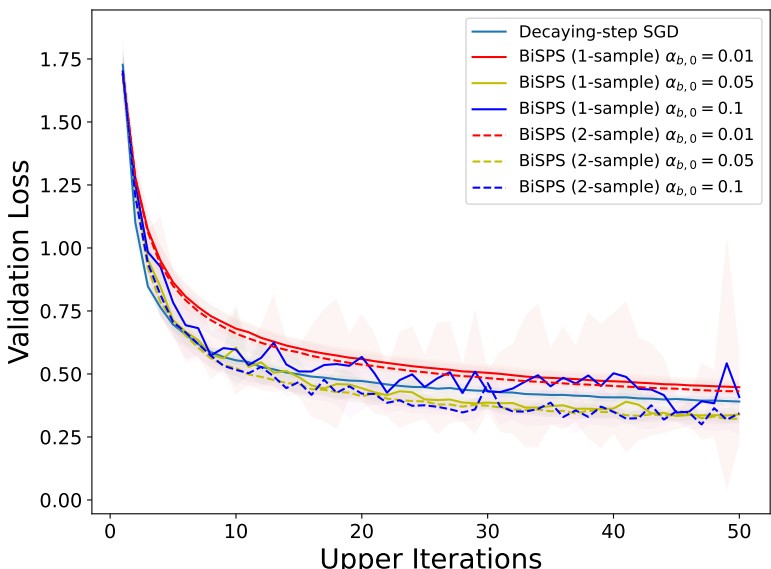

Figure 17: Comparison between BiSPS (2-samples), BiSPS (1-sample) and decaying-step SGD. Experiments are based on hyper-representation learning. For either version of BiSPS, the lower-level learning rate ($\beta_k$) is fixed at 10. The hypergradient is computed using conjugate gradient [17].

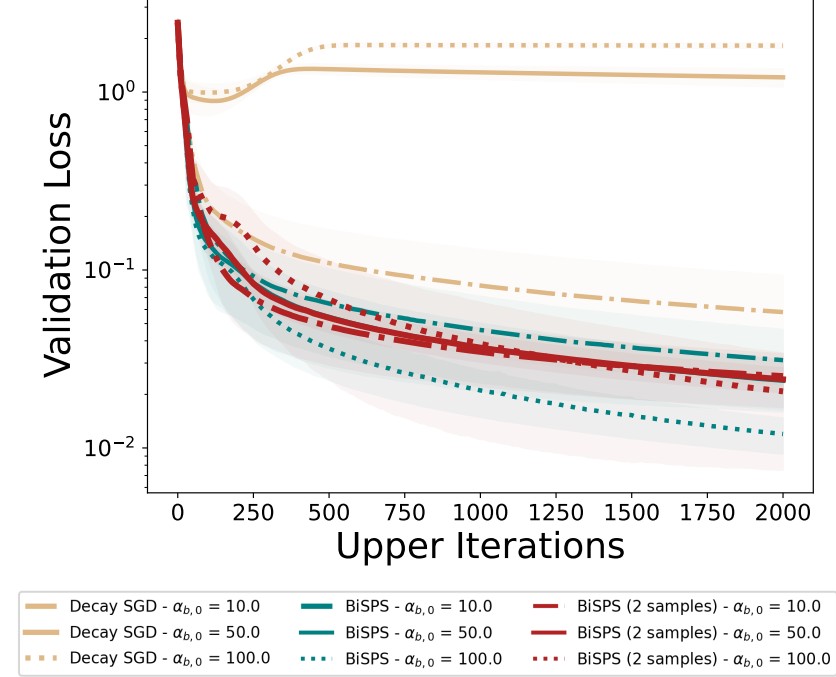

Figure 18: Comparison between BiSPS (2-samples), BiSPS (1-sample) and decaying-step SGD. Experiments are based on data distillation. For either version of BiSPS, the lower-level learning rate $(\beta_k)$ is fixed at $10^{-4}$. The Inverse Hessian in (2) is treated as the Identity when computing the hypergradient [30].

