# OpenReview forum: "BiSLS/SPS: Auto-tune Step Sizes for Stable Bi-level Optimization"
_NeurIPS.cc/2023/Conference — NeurIPS 2023 poster_

### Official Review · Reviewer_75AD · 2023-07-02

**Soundness:** 3 good
**Presentation:** 2 fair
**Contribution:** 2 fair
**Rating:** 5
**Confidence:** 4

**Summary:**

This paper studies the bilevel optimization problem, an in particular, focuses on developing effective learning rate schemes for bilevel optimization. In specific, the authors propose two adaptive step-sizes methods named stochastic line search (SLS) and stochastic Polyak step size (SPS) as variants of methods such as SPSmax and DecSPS. Compared to existing approaches, the proposed methods do not require the step-sizes to be monotonic by replacing the constant $\gamma_{b,0}$ by a non-increasing $\gamma_{b,k}$. The adaptive stepsizes are further applied to bilevel optimization on both upper and lower levels. The convergence analysis is provided for SPS and SLS on single-level problems and also for bilevel problems. Simple experiments are provided.


**Strengths:**

1. Studying adaptive learning rates in bilevel optimization is interesting, and has not been explored well in the literature.

2. The proposed adaptive learning rates are more practical than existing methods like SPSmax and DecSPS with milder requirements.

**Weaknesses:**

1. The paper is well written and hard to follow. For example, there are quite a few assumptions and requirements in the paragraphs of describing the proposed methods. There are also many notations and inequalities in Section 2. It makes me a little bit hard to follow the principle of the designs.

2. The adaptive learning rates seem to introduce more hyperparameters such as $l^*_{i_k}, c_{k},\gamma_{b,k}$. It makes me wonder how it can be meaningful in practice given more efforts in tuning such parameters. In addition, no sensitivity analysis is provided for such parameters in experiments.

3. How to find the approximate $l^*_{I,k}$ of the lowest function value? Is the choice of this quantity important in terms of performance? A more comprehensive empirical justification should be provided.

4. In terms of the complexity, it seems to me the proposed stepsizes cannot provide improved sample or computational complexity in theory. Maybe I miss something, but I think the authors can elaborate on this

5. The experiments are not convincing enough. Since the biggest motivation lies in the design of adaptive learning rates, the benefits may come from the empirical side. Thus, it would be better to provide more comprehensive experiments on practical NN architecture and larger datasets for the validation. However, the current experiments are rather toy examples. If the contribution lies in the theoretical side, it fails to show improved complexity performance.

Overall, I like the topic of adaptive learning rate for bilevel problems. However, the current theory and experiments are not convincing enough, and how important the proposed methods also are not clear to me. For this reason, I am on the negative side for the current version. I am also open to change my mind if more convincing experiments (e.g., how to select extra parameters, more datasets and backbones, sensitivity analysis, more problems) can be provided.

**Questions:**

Please see Weakness part.

**Limitations:**

Please see Weakness part.

---

> ### Author Rebuttal · Authors · 2023-08-07
>
> **Reply to weakness 1:** We would appreciate if the reviewer could elaborate on specific issues that might be unclear to them. This would enable us to address these concerns and enhance the overall clarity of our work.
>
> **Reply to weakness 2:** Let us clarify the roles of these hyperparameters. Starting with $l_{i_k}^*$, note this is simply a **lower bound** on the minimum function value. In particular, it can be taken as $0$ for positive losses such as cross-entropy and we set it to be $0$ for all experiments.  With regards to the parameters $c_k$ and $\gamma_{b,k}$, we emphasize that our algorithm is robust to these choices. To further illustrate this, we provide below additional experiments on the minimum train loss for different values of $\gamma_{b,0} \in$ \{$ 100,500,1000,2000$\} with $c_k = 1, \forall k$ (note the decay schedule for $\gamma_{b,k}$ is $\frac{\gamma_{b,0}}{\sqrt{k+1}}$).
>
> | $\gamma_{b,0}$ | 100 | 500 | 1000 | 2000 |
> | -------- | -------- | -------- | -| -|
> | Minimum Train Loss     |  0.04821 $\pm$ 2e-05 |  0.04458 $\pm$ 3e-05 | 0.04433 $\pm$ 5e-05 | 0.04475 $\pm$ 9e-05|
>
> We further show the results for different values of $c_k \in$ \{$1,2,5,10$\} with $\gamma_{b,0} = 1000$.
>
> | $c_k$ | 1|2|5|10|
> |- |- |-|-| -|
> | Minimum Train Loss|0.04433 $\pm$ 5e-05| 0.04439 $\pm$ 6e-05 | 0.04453 $\pm$ 6e-05| 0.04497 $\pm$ 5e-05|
>
> In particular, we wish to contrast these results to the sensitivity of decaying-step SGD to its learning rate. For the latter, the table below shows the minimum train loss of SGD changes much more drastically as the learning rate changes when compared against our algorithm (note the learning rate schedule is chosen to be $\frac{\gamma_{b,0}}{\sqrt{k+1}}$).
>
> | $\gamma_{b,0}$ | 100 | 500 | 1000 | 2000 |
> | -------- | -------- | -------- | -| -|
> | Minimum Train Loss     |  0.04945 $\pm$ 8e-05 |  0.073 $\pm$ 1e-03 | 0.119 $\pm$ 3e-03 | 0.220 $\pm$ 6e-03|
>
> **We did a thorough study on the extra parameters associated with our algorithmic design in the bi-level setting, specifically, search initializations for upper ($\alpha_{b,0}$) and lower-level learning rates ($\beta_{b,0}$) (Figure 6c, 6d), sensitivity of the algorithm to $\delta$ in eqn (14) (Figure 12, 13 in Appendix), performance of different reset options in Algorithm 2 (Figure 9 in Appendix), sensitivity of $\eta$ in reset option 3 (Figure 10 in Appendix), and upper and lower-level search cost (Figure 11 in Appendix). We want to emphasize that BiSLS is highly robust to the parameters associated with the algorithm, namely $\alpha_{b,0}$, $\beta_{b,0}$, $\delta$, and $\eta$** (note that reset option 3  has the best performance in terms of convergence speed, generalization, and computation cost). Besides this, we also observe that BiSPS is much more robust to $\alpha_{b,0}$ (upper-level learning rate bound) when compared against decaying-step SGD (Figure 4).
>
> **Reply to weakness 3:** see reply for weakness 2.
>
> **Reply to weakness 4:** The convergence rate matches the best rate of SGD [1]  while not requiring exhaustive step-size tuning.
>
> [1] Chen et al. Tighter Analysis of Alternating Stochastic Gradient Method for Stochastic Nested Problems
>
> **Reply to weakness 5:** **The key contribution of this work is to address the question of how to remove the extensive manual tuning of the two learning rates in bi-level optimization  (Figure 5b,5c)**. The experiments of hyper-representation learning and  data distillation are adapted from [2] and [3], respectively, using neural networks and real datasets. These are important and recent applications of bi-level optimization. The experiments in the bi-level setting are more challenging than the single-level setting as the computation of hypergradient typically involves second-order gradient information which can incur a high memory cost.
>
> [2] Sow et al. On the Convergence Theory for Hessian-Free Bilevel Algorithms
>
> [3] Lorraine et al. Optimizing Millions of Hyperparameters by Implicit Differentiation
>
> To concretely demonstrate the computation efficiency of our approach, we performed additional experiments on the run time of an algorithm to reach 85% validation accuracy for different number of Conjugate Gradient (CG) steps (results with units in seconds are given in the table below). We observe the consistent improvement of our algorithm over the baseline. Besides this, the baseline requires extensive tuning of the two learning rates which adds significant more computation cost at first place (not included in the table).
>
> | CG steps | 5| 10 | 15 | 20 |25|
> | -| -|-|-|-|-|
> | Adam| 138.4 $\pm$ 15.5 | 131.8 $\pm$ 14.7| 144.6 $\pm$ 23.6| 158.1 $\pm$ 19.6| 169.93 $\pm$ 22.4|
> |BiSLS-Adam (ours) |84.7 $\pm$ 14.5| 84.0 $\pm$ 10.5| 98.5 $\pm$ 20.9| 110.3 $\pm$ 19.5|117.6 $\pm$ 22.7|
>
> Furthermore, we added experiments that change the number of gradient steps (given in the top row of the table below) for approximating $y^*(x)$ when executing the line search steps according to eqn (14) (note this number is limited to be $1$ in eqn (14)). The results suggest that a single step already gives good performance. Further increasing this number does not lead to significant improvement considering the increase in the run-time of the algorithm.
>
> |  | 1| 2 | 5 | 7 |10|
> | -| -|-|-|-|-|
> | Best validation accuracy (units in 100%)| 91.42 $\pm$ 0.70 | 91.78 $\pm$ 0.70| 91.88 $\pm$ 0.83| 91.86 $\pm$ 0.99| 92.00 $\pm$ 0.95|
> |Time to reach 85% validation accuracy (units in seconds)| 110.10 $\pm$ 19.70| 113.76 $\pm$ 20.49|161.91 $\pm$ 26.85|181.74 $\pm$ 22.84|244.34 $\pm$ 38.02|
>
> Finally, our experiments demonstrate that the proposed algorithm can work with different types of hypergradient computation including CG (Figure 5a), Neumann series (Figure 7a), or the Hessian inverse being treated as identity (Figure 7b).
>
> We believe that our experiments are comprehensive. Thus, we are happy to discuss further if there is any remaining  concerns.

---

> > ### Comment · Reviewer_75AD · 2023-08-14
> > **Thanks for the response**
> >
> > I thank the authors' rebuttal, and I increase my score to 5.

---

### Official Review · Reviewer_ysEm · 2023-07-07

**Soundness:** 3 good
**Presentation:** 3 good
**Contribution:** 3 good
**Rating:** 7
**Confidence:** 3

**Summary:**

This paper presents an adaptive step size algorithm for bi-level optimization, which improves the shortcomings of the existing BO method that requires careful adjustment of the upper and lower learning rate, and gives a proof of convergence. Experiments also verify the robustness of the method to learning rate.

**Strengths:**

1.This paper presents an adaptive and robust bi-level optimization algorithm for adaptive step size, which can obtain a good set of step sizes without prior knowledge and careful modulation.
2.This algorithm is compatible with the accelerated solver Adam in addition to SGD, improving computational efficiency.
3.The analysis framework of this algorithm more generally unifies SPS and SLS.

**Weaknesses:**

1.Theorem 1 only states that f is convex rather than strongly convex, but the full text lacks explanation for the case of multiple solutions at the lower level
2.Potential computational burden. Although the author states that only a small number of matrix vector multiplication and addition operations are usually used to approximate matrix inverses, this still implies additional computational complexity and there is a problem: the trade-off between the performance loss caused by approximation and the improvement of computational efficiency. Specifically, the lack of time analysis in the charts in the paper exacerbates this concern.



**Questions:**

1.The drawing of  figure is difficult to read. For example, the line of beta=10.0 in Figure 1 is not fully drawn, different colors and lines are mixed in Figure 2 but have no additional meaning, and the second half of the right subgraph in Figure 3 is too complex, resulting in some lines being unrecognizable.
2. The light colored parts in the figure do not seem to represent the three standard deviations, but rather the upper and lower bounds?
3.I noticed that the sequence of upper and lower bounds for the declared step size in the paper needs to be appropriately controlled, so my point of interest is whether the decay rate is not 1/ sqrt {k+1}? Or can I change the decay rate to change the rate of convergence?

**Limitations:**

Yes, the author clearly stated the assumptions of the algorithm.

---

> ### Author Rebuttal · Authors · 2023-08-07
>
> Thank you for your positive assessment of our work.
>
> weakness 1: Theorem 1 only states that f is convex rather than strongly convex, but the full text lacks explanation for the case of multiple solutions at the lower level
>
> **Reply to weakness 1:**  Let us please clarify that Theorem 1 is for the case of single-level optimization; thus, there is no lower-level optimization.
>
> weakness 2: Potential computational burden. Although the author states that only a small number of matrix vector multiplication and addition operations are usually used to approximate matrix inverses, this still implies additional computational complexity and there is a problem: the trade-off between the performance loss caused by approximation and the improvement of computational efficiency. Specifically, the lack of time analysis in the charts in the paper exacerbates this concern.
>
> **Reply to weakness 2:** We want to emphasize that there is no additional backpropagation operations when executing the line search steps until eqn (14) is satisfied. Hence, the matrix vector multiplications are mainly associated with computing the hypergradient. Here, we have added additional experiments on the run-time (in seconds) of the algorithm to reach 85% validation accuracy for different number of Conjugate Gradient (CG) steps given in the table below. We observe the consistent improvement of our algorithm over the baseline in terms of computation time. This is due to the suitable and potentially large learning rates found by our algorithm (Figure 5b, 5c). Besides this, a more significant computation cost of the baseline is tuning the learning rates (not included in the table) which is not required by our algorithm.
>
> | CG steps | 5| 10 | 15 | 20 |25|
> | -| -|-|-|-|-|
> | Adam| 138.4 $\pm$ 15.5 | 131.8 $\pm$ 14.7| 144.6 $\pm$ 23.6| 158.1 $\pm$ 19.6| 169.93 $\pm$ 22.4|
> |BiSLS-Adam (ours)|84.7 $\pm$ 14.5| 84.0 $\pm$ 10.5| 98.5 $\pm$ 20.9| 110.3 $\pm$ 19.5|117.6 $\pm$ 22.7|
>
> To further explore the trade-off between performance loss and computation efficiency, we have added experiments that change the number of steps for approximating $y^*(x)$ in eqn (14) (note that we limit it to be 1 in eqn (14)). We observe that increasing this number can improve the performance of the algorithm, but the gain may not be significant considering the extra overhead introduced, e.g., comparing the number of steps being 1 and 10. Figure 11 in Appendix provides additional information on the search cost for different values of $\eta$ in reset option 3.
>
> |  | 1| 2 | 5 | 7 |10|
> | -| -|-|-|-|-|
> | Best validation accuracy (units in 100%)| 91.42 $\pm$ 0.70 | 91.78 $\pm$ 0.70| 91.88 $\pm$ 0.83| 91.86 $\pm$ 0.99| 92.00 $\pm$ 0.95|
> |Time to reach 85% validation accuracy (units in seconds)| 110.10 $\pm$ 19.70| 113.76 $\pm$ 20.49|161.91 $\pm$ 26.85|181.74 $\pm$ 22.84|244.34 $\pm$ 38.02|
>
> Questions: 1.The drawing of figure is difficult to read. For example, the line of beta=10.0 in Figure 1 is not fully drawn, different colors and lines are mixed in Figure 2 but have no additional meaning, and the second half of the right subgraph in Figure 3 is too complex, resulting in some lines being unrecognizable. 2. The light colored parts in the figure do not seem to represent the three standard deviations, but rather the upper and lower bounds? 3.I noticed that the sequence of upper and lower bounds for the declared step size in the paper needs to be appropriately controlled, so my point of interest is whether the decay rate is not 1/ sqrt {k+1}? Or can I change the decay rate to change the rate of convergence?
>
> **Reply to Q1:** Thanks for the suggestions regarding Figures 2 and 3. We will improve their readability in the revision. For $\beta=10.0$, the algorithm diverges after the period in which the line is drawn.
> **Reply to Q2:** The light colored parts are standard deviations.
>
> **Reply to Q3:** We expect that using different envelopes can give faster rates under additional assumptions (or achieve similar rates in more-general settings).

---

> > ### Comment · Reviewer_ysEm · 2023-08-19
> >
> > Thanks for the author's rebuttal. It solved my concerns. I don't have any other questions at the moment.

---

### Official Review · Reviewer_evaJ · 2023-07-07

**Soundness:** 4 excellent
**Presentation:** 4 excellent
**Contribution:** 4 excellent
**Rating:** 8
**Confidence:** 4

**Summary:**

This work studies adaptive step-size methods for both single-level optimization and bilevel optimization. The authors propose two novel variants of stochastic line search (SLS) and stochastic Polyak step size (SPS), and they unify these variants into a general envelop strategy. Importantly, these variants are simpler to implement and demonstrate good empirical performance, particularly in non-interpolating scenario. By using the unified envelop strategy, the authors also propose a bi-level line-search algorithm BiSLS-Adam/SGD with convergence guarantees, which demonstrates empirical robustness and generalizes well.

**Strengths:**

1. Both adaptive step-size methods and bilevel optimization are currently active topics. The investigation of auto-tune step sizes for bilevel optimization algorithms is under-explored, and the studied topic in work is interesting and important.
2. The illustrations in Figures 1-5 are helpful in understanding the contributions.
3. The newly proposed variants of stochastic line search (SLS) and stochastic Polyak step size (SPS) in this work are novel and easy to understand. Moreover, these variants can be unified into a general envelop-type step size, and their effectiveness in the context of single-level optimization and bilevel optimization is well-supported by theoretical results and extensive experiments.

**Weaknesses:**

See the Limitations below.

**Questions:**

Q1: Can the authors explain in more details why it is possible to set the number of inner steps to be 1 in Line 213?

Minor Comments:
1. For Equation (2), the $+$ should be $-$.

2. In Equation (13), $\nabla_y^2 yg$ should be $\nabla_{yy}^2 g$.

3. In Line 175, $\nabla f_x$ should be $\nabla_x f$.

4. The quadratic functions in Figure 2 and Section B.1 do not satisfy the gradient bounded assumption in Assumption 2.

5. The $+$ in Equation (17) should be $-$.

**Limitations:**

The authors discussed some of the limitations on single-level optimization in the conclusion section of the paper.

On bilevel optimization, a limitation is the lower-level strong convexity in Assumption 3. It would be of interest to investigate whether this condition can be relaxed or removed, taking into account recent advancements in the field, such as those presented in [1, 2,3].

[1] B. Liu et al. “Bome! bilevel optimization made easy: A simple first-order approach.” NeurIPS 2022.

[2] R. Liu et al. “Averaged Method of Multipliers for Bi-Level Optimization without Lower-Level Strong Convexity.” ICML 2023.

[3] H. Shen and T. Chen, “On penalty-based bilevel gradient descent method.” ICML 2023.

---

> ### Author Rebuttal · Authors · 2023-08-07
>
> Thank you for your positive feedback!
>
> We appreciate your careful reading of our paper and thank you for pointing out a few typos, which will be fixed in the revision.
>
> Q1: Can the authors explain in more details why it is possible to set the number of inner steps to be 1 in Line 213?
>
> **Reply to Q1:** Although we have not formally analyzed this heuristic, we have found that it works well empirically while significantly decrease the computation cost. Specifically, while more steps can be used to give a better approximation of $y^*(x)$ in the nested loop, they don't seem to give significant improvements in terms of validation accuracy considering the extra overhead introduced. We have added additional experiments to justify this point with changing number of steps for approximating $y^*(x)$ given in the top row of the table below.
>
> |  | 1| 2 | 5 | 7 |10|
> | -| -|-|-|-|-|
> | Best validation accuracy (units in 100%)| 91.42 $\pm$ 0.70 | 91.78 $\pm$ 0.70| 91.88 $\pm$ 0.83| 91.86 $\pm$ 0.99| 92.00 $\pm$ 0.95|
> |Time to reach 85% validation accuracy (units in seconds)| 110.10 $\pm$ 19.70| 113.76 $\pm$ 20.49|161.91 $\pm$ 26.85|181.74 $\pm$ 22.84|244.34 $\pm$ 38.02|
>
> Limitations: The authors discussed some of the limitations on single-level optimization in the conclusion section of the paper. On bilevel optimization, a limitation is the lower-level strong convexity in Assumption 3. It would be of interest to investigate whether this condition can be relaxed or removed, taking into account recent advancements in the field, such as those presented in [1, 2, 3].
>
> **Reply to Limitations:** Thank you for highlighting these recent works. While lower-level strong-convexity is a standard assumption in bi-level optimization, e.g. [4,5,6,7], we are eager to study in detail the references that you mentioned hoping to extend our results to that context in future research.
>
> [4] Ghadimi and Wang Approximation Methods for Bilevel Programming
>
> [5] Hong et al. A Two-Timescale Framework for Bilevel Optimization: Complexity Analysis and Application to Actor-Critic
>
> [6] Ji et al. Bilevel Optimization: Convergence Analysis and Enhanced Design
>
> [7] Chen et al. Tighter Analysis of Alternating Stochastic Gradient Method for Stochastic Nested Problems

---

> > ### Comment · Reviewer_evaJ · 2023-08-15
> >
> > Thanks for the rebuttal and I do not have further question for the moment.

---

### Official Review · Reviewer_TsPX · 2023-07-07

**Soundness:** 3 good
**Presentation:** 3 good
**Contribution:** 3 good
**Rating:** 6
**Confidence:** 3

**Summary:**

The paper introduces the use of stochastic adaptive-step size methods, namely stochastic Polyak step size (SPS) and stochastic line search (SLS), for bi-level optimization. This approach addresses the challenge of tuning both the lower and upper-level learning rates.

**Strengths:**

1. SLS and SPS can be seen as special instances of general family of methods with an envelope-type step-size.
2. The unified envelope strategy enables the algorithm development and convergence analysis.

**Weaknesses:**

The paper compares the proposed algorithms to vanilla SGD or Adam versions, but it does not provide a comprehensive comparison with other existing algorithms for bi-level optimization. This may limit the understanding of the relative performance of the proposed algorithms in the broader context.

**Questions:**

What is the overhead of using the proposed algorithm to tune the step size?

---

> ### Author Rebuttal · Authors · 2023-08-07
>
> Thank you for your overall positive assessment of our work.
>
> weakness: The paper compares the proposed algorithms to vanilla SGD or Adam versions, but it does not provide a comprehensive comparison with other existing algorithms for bi-level optimization. This may limit the understanding of the relative performance of the proposed algorithms in the broader context
>
> **Reply to weakness:** We would greatly appreciate if the reviewer could provide more specific details regarding particular methods they have in mind that could potentially be missing for comparisons. The primary focus of our work centers around the intricacies of tuning the two learning rates within the context of bi-level optimization. Thus, we try to make the comparison as straight and fair as possible under the alternating optimization framework given in eqn (3). For instance, some bi-level optimization algorithms that rely on variance reduction or momentum may require additional hyperparameter tuning, which falls out of the scope of our work. Nonetheless, we believe that our approach can be integrated with these methods, since the general motivation behind line-search is to find the learning rate given a suitable direction. Moreover, we have already shown the compatibility of our algorithm with various techniques for computing the hypergradient, such as Conjugate Gradient, Neumann Series, or the Hessian inverse being treated as identity (Figure 5a, 7a, 7b).
>
> question: What is the overhead of using the proposed algorithm to tune the step size?
>
> **Reply to question:** The overhead is the execution of line search steps until the modified Armijo line-search rule given in eqn 14 is satisfied. To avoid always searching from initial learning rates $\alpha_{b,0}$ and $\beta_{b,0}$, we have come up with a reset subroutine (Algorithm 2) that sets the search starting point of current iteration to be a factor $\eta$ times the previous iteration's learning rate (option 3). Its full descriptions can be found in Section 2 and Appendix B.2. We have shown that our algorithm is robust to the choice of $\eta$ in Figure 10 in Appendix. Furthermore, Figure 11 in Appendix shows the cost for finding the upper and lower-level learning rates are $9$ and $1$ respectively when $\eta = 2$ (measured in terms of average number of search rounds until eqn (14) is satisfied per iteration).
>
> To further address your question, we added additional experiments on the run-time of the algorithm measured in seconds. Despite the search cost, our algorithm reaches a threshold validation accuracy (85%) faster than the baseline for different number of Conjugate Gradient (CG) steps as shown in the table below. This is because our algorithm is able to find suitable and potentially large learning rates as demonstrated in Figure 5b and 5c. Moreover, tuning the baseline learning rates has a significant higher computation cost (not included in the table) than running the algorithm itself, which is resolved by our proposed approach.
>
> We are happy to discuss further if there are any additional questions.
>
> | CG steps | 5| 10 | 15 | 20 |25|
> | -| -|-|-|-|-|
> | Adam| 138.4 $\pm$ 15.5 | 131.8 $\pm$ 14.7| 144.6 $\pm$ 23.6| 158.1 $\pm$ 19.6| 169.93 $\pm$ 22.4|
> |BiSLS-Adam (ours)|84.7 $\pm$ 14.5| 84.0 $\pm$ 10.5| 98.5 $\pm$ 20.9| 110.3 $\pm$ 19.5|117.6 $\pm$ 22.7|

---

### Author Rebuttal · Authors · 2023-08-07

Dear Reviewers,

We thank you for taking the time to carefully read and review our work. We are confident that integrating your suggestions will further improve our paper; thus we plan on doing so in the revision.
In the specific replies below, we comment on specific issues brought up by different reviewers.

We are open to further discussions should any additional questions arise.

---

### Decision · Program_Chairs · 2023-09-21

**Decision:**

Accept (poster)

**Comment:**

The reviewers are in consensus that this work is novel and significant, unifying the Polyak step size and stochastic line search in a common framework for bilevel optimization. The authors addressed reviewers' concerns comprehensively, including running additional experiments to resolve some reviewer curiosities. The authors are encouraged to incorporate the reviewers' remaining suggestions, which arose during the discussion period.